# ImageDoctor: Diagnosing Text-to-Image Generation via Grounded Image Reasoning

**Yuxiang Guo**[1,2*] **Jiang Liu**[2*†] **Ze Wang**[2] **Hao Chen**[2] **Ximeng Sun**[2]
**Yang Zhao**[1] **Jialian Wu**[2] **Xiaodong Yu**[2] **Zicheng Liu**[2] **Emad Barsoum**[2]
[1]Johns Hopkins University  [2]AMD
https://image-doctor.github.io/

## Abstract

The rapid advancement of text-to-image (T2I) models has increased the need for reliable human preference modeling, a demand further amplified by recent progress in reinforcement learning for preference alignment. However, existing approaches typically quantify the quality of a generated image using a single scalar, limiting their ability to provide comprehensive and interpretable feedback on image quality. To address this, we introduce ImageDoctor, a unified multi-aspect T2I model evaluation framework that assesses image quality across four complementary dimensions: plausibility, semantic alignment, aesthetics, and overall quality. ImageDoctor also provides pixel-level flaw indicators in the form of heatmaps, which highlight misaligned or implausible regions, and can be used as a dense reward for T2I model preference alignment. Inspired by the diagnostic process, we improve the detail sensitivity and reasoning capability of ImageDoctor by introducing a *"look-think-predict"* paradigm, where the model first localizes potential flaws, then generates reasoning, and finally concludes the evaluation with quantitative scores. Built on top of a vision-language model and trained through a combination of supervised fine-tuning and reinforcement learning, ImageDoctor demonstrates strong alignment with human preference across multiple datasets, establishing its effectiveness as an evaluation metric. Furthermore, when used as a reward model for preference tuning, ImageDoctor significantly improves generation quality—achieving an improvement of 10% over scalar-based reward models.

## 1 Introduction

With the rapid evolution of text-to-image (T2I) architectures (Goodfellow et al., 2014; Croitoru et al., 2023; Gu et al., 2023; Xie et al., 2024; Tian et al., 2024; Wang et al., 2025c), the quality of generated images has advanced significantly. Modern T2I systems can now produce highly realistic outputs that closely follow textual instructions, enabling a wide range of applications in areas such as art, design, and entertainment. These advances, however, give rise to a critical question: *how to reliably evaluate the quality of generated images that may suffer from poor instruction adherence, low aesthetic quality, or counterintuitive artifacts*. Furthermore, with the rise of reinforcement learning (Liu et al., 2025; Xue et al., 2025) and test-time scaling (Guo et al., 2025; Ma et al., 2025a), evaluators play an increasingly important role: not only can they serve as reward functions or verifiers to measure quality, but they are also expected to provide actionable feedback to improve generation.

Current human preference models, such as HPS (Wu et al., 2023b), ImageReward (Xu et al., 2023), and PickScore (Kirstain et al., 2023), mostly predict a single scalar score of quality. However, compressing the evaluation into a single scalar is often insufficient to capture the detailed flaws of the generated images. For instance, two images may receive the same score, yet differ substantially: one may be aesthetically pleasing but poorly aligned with the prompt, while the other may follow the prompt faithfully but contain unrealistic artifacts. Relying solely on a single score cannot disentangle these factors, limiting both the interpretability and the usefulness of the feedback for guiding model improvement. Moreover, existing evaluators can only provide an overall judgment of image quality but lack spatially grounded feedback, *i.e.*, they cannot identify *where* in the image

---

[*]Equal Contribution    [†]Corresponding Author

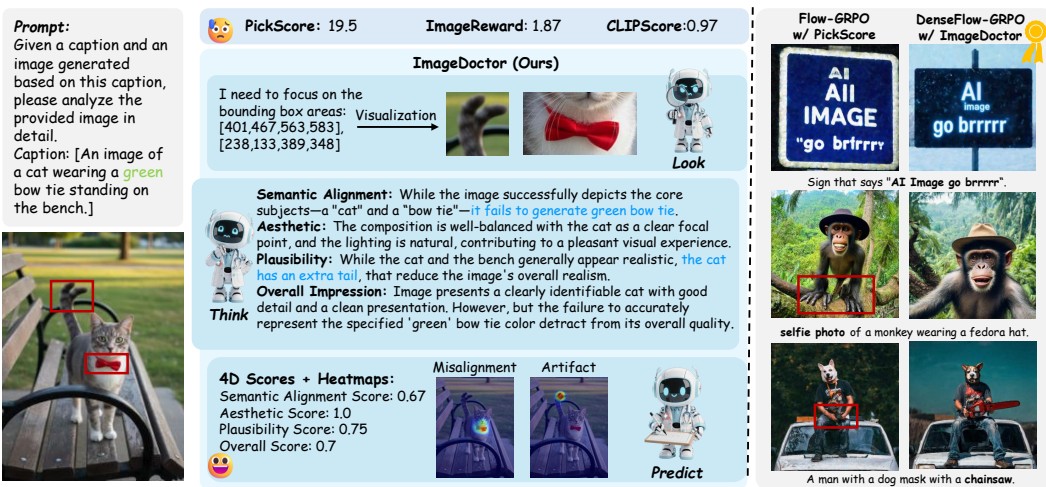

Figure 1: **Comparison between ImageDoctor and scalar-based reward functions.** *Left*: ImageDoctor follows a *"look-think-predict"* paradigm, providing rich feedback with four-dimensional scores and heatmaps that highlight misalignment and artifact locations. *Right*: Leveraging this fine-grained feedback, DenseFlow-GRPO (Sec. 4) generates images with more faithful and realistic local details, outperforming Flow-GRPO, which relies on the scalar-based reward PickScore.

the problems occur. In practice, many T2I failures stem from partial prompt adherence: while the majority of the prompt may be satisfied, fine-grained details are often missing or incorrect. This absence of localization further reduces the interpretability and actionability of these evaluators, especially when used as reward functions. These challenges highlight the need for evaluators that can provide comprehensive feedback, offering both multi-dimensional quality scores and localized diagnostics—much like a doctor diagnosing the problems in an image.

In this work, we propose **ImageDoctor**, a unified evaluation framework that produces holistic scoring and spatially grounded feedback in the form of artifact and misalignment heatmaps. Steering the reasoning strengths and commonsense knowledge of multi-modal large language models (MLLMs), ImageDoctor is built on a fine-tuned MLLM backbone to achieve a deep joint understanding of images and prompts. To achieve flaw localization, we introduce a lightweight heatmap decoder that produces the heatmaps highlighting misalignment and artifact locations conditioned on the input prompt, image features, and the response generated by the MLLM. Inspired by the process of medical diagnosis, we further propose a *"look-think-predict"* paradigm as shown in Fig. 1. Before final judgment, ImageDoctor performs grounded image reasoning, which consists of two steps. First, it pinpoints potential flawed regions that require closer attention in the image (*"look"*). Then, it analyzes these regions by integrating the localized visual evidence with contextual understanding, generating structured reasoning that evaluates the image from multiple aspects (*"think"*). We incentivize this grounded image reasoning capability with cold start and reinforcement finetuning.

Reinforcement learning from human feedback (RLHF) (Xue et al., 2025; Liu et al., 2025) has been proven effective in enhancing both image quality and text-image alignment for T2I generation. Nevertheless, current RLHF approaches, such as Flow-GRPO (Liu et al., 2025), rely solely on sparse *image-level* rewards, which overlook spatially localized feedback and thus fail to provide fine-grained guidance during training. To address this limitation, we introduce **DenseFlow-GRPO**, a new RLHF framework that enhances T2I models with both *image-level* and *pixel-level* dense reward signals. By leveraging the rich diagnostic feedback from ImageDoctor, DenseFlow-GRPO delivers more precise and spatially aligned supervision, enabling T2I models to learn not only what constitutes a good image globally, but also how to refine local regions in a fine-grained manner.

Our experiments demonstrate that ImageDoctor achieves state-of-the-art alignment with human judgments, substantially improving score prediction accuracy across all dimensions (average PLCC 0.741 vs. 0.586 of the previous best on RichHF-18K). Beyond serving as a metric, ImageDoctor generalizes well to downstream applications: as a verifier, it reliably selects higher-quality generations in test-time scaling; as a reward model, it drives consistent gains in reinforcement learning. In particular, integrating ImageDoctor into Flow-GRPO yields superior preference alignment, and

further utilizing the dense heatmap feedback in DenseFlow-GRPO achieves the strongest improvements and delivers images with more faithful local details.

The main contributions of this paper are summarized below:

- We propose ImageDoctor, a unified T2I evaluation model that produces dense feedback, including multi-aspect scores and heatmaps localizing flaws, enabling interpretable and fine-grained assessment.

- We introduce a *"look-think-predict"* paradigm that equips ImageDoctor with structured reasoning by integrating visual grounding and textual analysis. ImageDoctor is further refined through reinforcement finetuning with tailored reward functions, enhancing adherence to human preferences while ensuring spatially grounded reasoning.

- We present DenseFlow-GRPO, a novel T2I reinforcement learning method that incorporates ImageDoctor's dense spatial feedback into the reward signal, providing region-aware supervision and leading to more robust improvements in image generation.

- Extensive experiments on human preference datasets demonstrate the ImageDoctor's superior alignment with human preference, and we further validate its effectiveness by applying it to downstream tasks, serving as a verifier and a reward model.

## 2 RELATED WORK

### 2.1 TEXT-TO-IMAGE GENERATION

Text-to-image (T2I) generation is a core task in generative modeling. It aims to synthesize semantically aligned images from natural language prompts, while balance the aesthetic quality and plausibility. Early approaches based on Generative Adversarial Networks (GANs) (Goodfellow et al., 2014) and Variational Auto-Encoders (VAEs) (Kingma & Welling, 2013) demonstrated feasibility but were limited by low diversity and coarse details. More recently, diffusion models (Ho et al., 2020; Nichol et al., 2021; Rombach et al., 2022) have emerged as a dominant paradigm, achieving significant gains in image quality and diversity. Flow-based models (Zhao et al., 2024; Esser et al., 2024; Labs, 2024) provide another class of likelihood-based generative models, relying on stochastic denoising steps, thereby enabling efficient sampling and reducing inference overhead. In addition, autoregressive models (Tian et al., 2024; Xie et al., 2024) are gaining attention for their compositionality and controllability, bridging vision and language more effectively.

### 2.2 T2I EVALUATION MODELS

With the rapid progress in T2I generation, evaluation models have also advanced, though the task remains highly challenging. CLIPScore (Hessel et al., 2021) was one of the earliest automatic metrics that leverages pretrained CLIP to compute the similarity between the generated image and its prompt. PickScore (Kirstain et al., 2023) and ImageReward (Xu et al., 2023) fine-tune CLIP (Radford et al., 2021) and BLIP (Li et al., 2022), respectively, on large-scale human preference datasets, significantly improving alignment with subjective human judgments. The Human Preference Score (HPS) series (Wu et al., 2023b;a; Ma et al., 2025b) expand the scale of annotations to enhance preference alignment. In particular, HPSv3 (Ma et al., 2025b) leverages an MLLM backbone to capture rich semantic representations from images and captions and incorporates uncertainty-aware ranking to learn human preferences from paired comparisons. ICT-HP (Ba et al., 2025) proposes the Image-Contained-Text (ICT) and High-Preference (HP) models by finetuning a CLIP model to evaluate both text-image alignment and the image aesthetics and detail quality. While both HPSv3 and ICT-HP demonstrate promising results in predicting image-level scalar scores, they do not provide localized evaluation and offer no interpretability regarding why an image is judged as good or bad. Recently, UnifiedReward-think (Wang et al., 2025a) and VisualQuality-R1 (Tian et al., 2024) explored reinforcement learning for evaluation model training for image quality score prediction. RichHF (Liang et al., 2024) and HELM (Lee et al., 2023) attempt to broaden evaluation by considering multiple dimensions, moving beyond single-score preference modeling.

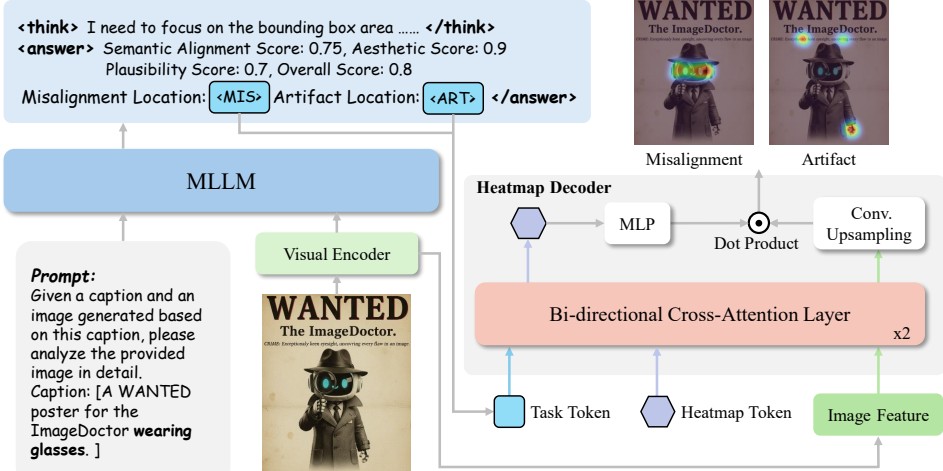

Figure 2: **ImageDoctor architecture.** Given a prompt-image pair, the MLLM follows a *"look-think-predict"* paradigm for T2I evaluation by localizing potential flaw regions, analyzing them, and generating holistic scores and special task tokens. The task token, with a learned heatmap token and image features are fed into the heatmap decoder to produce the misalignment and artifact heatmaps.

## 3    IMAGEDOCTOR

ImageDoctor aims to provide rich, interpretable, and accurate diagnoses for T2I generation. To this end, we design a novel unified model architecture to generate both *image-level* scores and *pixel-level* heatmap evaluations leveraging the strong image and text understanding capabilities of multimodal large language models (MLLMs) (Sec. 3.1). To generate interpretable and accurate evaluations, we propose a *"look-think-predict"* paradigm, where the model first identifies possible local flaw regions and generates explicit reasoning about image details before providing final evaluations (Sec. 3.2).

### 3.1    MODEL DESIGN

**Overview.**    The overall pipeline of ImageDoctor is shown in Fig. 2. The input text prompt $P$ and the corresponding image $I \in \mathbb{R}^{H \times W}$ are passed into the MLLM backbone. ImageDoctor reasons about image details and image-text semantics to produce both holistic scores and localized diagnostic signals. Specifically, it outputs four scalar scores, *i.e.*, semantic alignment, aesthetics, plausibility, and overall scores $s_d, \forall d \in \{\text{align, aesth, plau, over}\}$, which evaluate image quality from different aspects. In addition, it provides localized feedback by marking image regions that are implausible or misaligned with the text through the artifact and misalignment heatmaps $H_d \in \mathbb{R}^{H \times W}, \forall d \in \{\text{art, mis}\}$ generated by the heatmap decoder.

**Heatmap Decoder.**    While scalar scores can be directly predicted via the text output of the MLLM backbone, generating pixel-wise heatmaps requires a unified model that supports both text and image outputs. To enable this, we design a lightweight heatmap decoder. The decoder takes the image features extracted by the visual encoder together with a learned heatmap token and a task token $\mathbf{t} \in \{\texttt{<ART>}, \texttt{<MIS>}\}$ representing the artifact and the misalignment tokens, respectively. The task tokens are generated by the MLLM backbone and fused with the input image, text prompt, and reasoning chains to guide accurate heatmap prediction. Inspired by the SAM mask decoder (Kirillov et al., 2023), we adopt a bi-directional cross-attention design to fuse the token and image embeddings. The updated image features are passed through a series of convolution upsampling layers to upscale to the original image size, from which the updated heatmap token are used to dynamically predict the heatmap. The detailed architecture of the heatmap decoder can be found in Section A.

### 3.2    GROUNDED IMAGE REASONING

As shown in Fig. 1, ImageDoctor adopts a *"look-think-predict"* paradigm to generate its evaluations for a given image and text prompt. Instead of directly generating the final prediction, it first

localizes potential flaw regions by predicting the flaw region bounding boxes (***"look"***), then analyzes and reasons about these flaws and overall image quality (***"think"***), and finally produces a conclusive judgment (***"predict"***), mimicking the image evaluation process of human. To enable the grounded image reasoning capability, we design a two-phase training pipeline. In the cold start phase, we conduct supervised fine-tuning to teach the model to predict image scores and heatmaps in the *"look-think-predict"* reasoning format. In the second phase, ImageDoctor adopts online reinforcement fine-tuning with Group Relative Policy Optimization (GRPO) (Shao et al., 2024) to further incentivize the reasoning ability to generate rich and reliable image feedback.

### 3.2.1 COLD START

Since MLLMs are designed for general image understanding, in the first cold start stage, we first finetune the MLLM backbone on the image evaluation task by training the model to directly predict the image evaluation scores. In the second cold start stage, we train ImageDoctor on chain-of-thought (CoT) data to learn the *"look-think-predict"* reasoning process for both score and heatmap predictions. To prepare the CoT data, we first detect the highlighted regions in the ground-truth artifact and misalignment heatmaps to generate flaw region bounding boxes. Then, we employ Gemini 2.5 Flash (Comanici et al., 2025) with carefully designed prompts to produce detailed reasoning between the image and corresponding human annotations. Finally, we organize the bounding boxes, reasoning traces, and ground-truth human annotations into the *"look-think-predict"* CoT format. The details can be found in Section B. The ImageDoctor model $\theta$ is optimized by:

$$\mathcal{L} = -\sum_i \log p_\theta(\boldsymbol{z}_i \mid \boldsymbol{z}_{<i}, \boldsymbol{I}, \boldsymbol{P}) + \sum_d \|\boldsymbol{H}_d - \tilde{\boldsymbol{H}}_d\|_2^2, \tag{1}$$

where $\boldsymbol{z}$ is the CoT reasoning text, $\boldsymbol{I}$ and $\boldsymbol{P}$ are the input image and text prompts, $\boldsymbol{H}_d$ and $\tilde{\boldsymbol{H}}_d$ are the ground-truth and predicted heatmaps, respectively.

### 3.2.2 REINFORCEMENT FINETUNING

After cold start, we further perform reinforcement finetuning (RFT) with GRPO (Shao et al., 2024) to enhance the reasoning ability of ImageDoctor. Given a pair of input $(\boldsymbol{I}, \boldsymbol{P})$, ImageDoctor as the policy model $\pi_\theta$, generates a group of $N$ candidate responses $\{\boldsymbol{o}^1, \ldots, \boldsymbol{o}^N\}$. For each response $\boldsymbol{o}^i$, we compute a reward score $\mathcal{R}^i$ using a combination of reward functions. The rewards are normalized within the group to compute the group-normalized advantage. RFT allows the model to explore diverse reasoning paths, directing it toward reasoning trajectories with high reward signals and enhancing its generalization capability. The detailed GRPO formulation can be found in Section C.

We design a suite of verifiable rewards to encourage the model to focus on the correct flaw regions, produce accurate evaluation scores, and generate precise heatmaps, including a grounding reward ($\mathcal{R}_G$), a score reward ($\mathcal{R}_S$) and a heatmap reward ($\mathcal{R}_H$).

**Grounding Reward** ($\mathcal{R}_G$) aims to evaluate whether the model can accurately locate the flaw regions in an image. The model should ideally generate a compact set of bounding boxes, both in number and area, that effectively cover the potential flaw regions. The grounding reward $\mathcal{R}_G$ has three complementary components: **1) Completeness.** The union of all bounding boxes should adequately cover the entire highlighted area in the artifact and misalignment heatmaps. We compute the ratio between the area covered by the union of all bounding boxes and the total intensity of the heatmaps. **2) Compactness.** Each bounding box should only cover flaw regions with minimal normal regions. We compute the average heatmap intensity within each predicted bounding box and then take the mean across all boxes, yielding higher rewards for bounding boxes with less normal regions. **3) Uniqueness.** The model should not predict redundant bounding boxes, and thus the overlap between any pair of boxes should be minimized. We measure the Intersection over Union (IoU) between each pair of bounding boxes and apply a penalty for large overlaps. Implementation details are in Section D.

**Score Reward** ($\mathcal{R}_s$) evaluates how well the predicted scores $\tilde{s}_d$ align with ground-truth human scores $s_d$. We use $\ell_1$ distance and design $\mathcal{R}_S = \sum_d 1 - \|s_d - \tilde{s}_d\|_1$, which encourages the model to produce score predictions that are close to human judgments.

**Heatmap Reward** ($\mathcal{R}_H$) measures the similarity between predicted heatmaps $\tilde{\boldsymbol{H}}_d$ and human annotated heatmaps $\boldsymbol{H}_d$. We use $\ell_2$ distance and define $\mathcal{R}_H = \sum_d 1 - \|\boldsymbol{H}_d - \tilde{\boldsymbol{H}}_d\|_2^2$. This formulation

assigns higher rewards when the predicted maps closely match the annotations, thereby encouraging the model to produce precise and sharp flaw localization heatmaps.

The total reward is the combination of the three rewards: $\mathcal{R} = \mathcal{R}_G + \mathcal{R}_S + \mathcal{R}_H$.

# 4 DENSEFLOW-GRPO: IMAGEDOCTOR AS DENSE REWARD

Reinforcement learning from human feedback (RLHF) (Xue et al., 2025; Liu et al., 2025) has demonstrated great success in improving image quality and image-text alignment for T2I generation. However, existing RLHF methods such as Flow-GRPO (Liu et al., 2025) adopt an *image-level* formulation without fine-grained supervision. Specifically, given a prompt $\boldsymbol{c}$, the flow model $p_\phi$ samples a group of $G$ individual images $\{\boldsymbol{x}_0^i\}_{i=1}^G$ and the corresponding reverse-time trajectories $\{(\boldsymbol{x}_T^i, \boldsymbol{x}_{T-1}^i, \cdots, \boldsymbol{x}_0^i)\}_{i=1}^G$. Flow-GRPO optimizes the flow model by maximizing the following:

$$\mathcal{J}_{\text{Flow-GRPO}}(\phi) = \frac{1}{G} \sum_{i=1}^{G} \frac{1}{T} \sum_{t=0}^{T-1} \left( \min\left(r_t^i(\phi)\hat{A}_t^i, \, \text{clip}\left(r_t^i(\phi), 1-\varepsilon, 1+\varepsilon\right)\hat{A}_t^i\right) - \beta D_{\text{KL}}(p_\phi||p_{\phi_{\text{ref}}}) \right),$$
(2)

where the likelihood ratio $r_t^i(\phi)$ and normalized advantage $\hat{A}_t^i$ of the $i$-th image are computed as:

$$r_t^i(\phi) = \frac{p_\phi(\boldsymbol{x}_{t-1}^i \mid \boldsymbol{x}_t^i, \boldsymbol{c})}{p_{\phi_{\text{old}}}(\boldsymbol{x}_{t-1}^i \mid \boldsymbol{x}_t^i, \boldsymbol{c})}, \quad \hat{A}_t^i = \frac{R(\boldsymbol{x}_0^i, \boldsymbol{c}) - \text{mean}(\{R(\boldsymbol{x}_0^i, \boldsymbol{c})\}_{i=1}^G)}{\text{std}(\{R(\boldsymbol{x}_0^i, \boldsymbol{c})\}_{i=1}^G)}.$$
(3)

In Eq. (3), both $r_t^i(\phi)$ and reward $R$ are computed on the image level. The reward signal is applied uniformly across all pixels in the image, treating every region equally, regardless of its quality. Finer-grained supervision is more desirable, as it allows low-quality regions to be penalized more, while encouraging high-quality areas. To fill this gap, we propose **DenseFlow-GRPO**, which enables both *image-level* and *pixel-level* fine-grained dense reward signals for flow model RL training, leveraging the rich image feedback generated by ImageDoctor. We first reformulate the likelihood ratio at each trajectory step to allow pixel-wise advantage customization:

$$s_t^i(\phi, h, w) = \text{sg}\left[r_t^i(\phi)\right] \cdot \frac{p_\phi(\boldsymbol{x}_{t-1}^i \mid \boldsymbol{x}_t^i, \boldsymbol{c})_{h,w}}{\text{sg}\left[p_\phi(\boldsymbol{x}_{t-1}^i \mid \boldsymbol{x}_t^i, \boldsymbol{c})_{h,w}\right]},$$
(4)

where $p_\phi(\boldsymbol{x}_{t-1}^i \mid \boldsymbol{x}_t^i, \boldsymbol{c})_{h,w}$ is the pixel-wise likelihood, $h, w$ denote the pixel location, and $\text{sg}[\cdot]$ is the stop-gradient operation that only takes the numerical value, corresponding to `detach` in PyTorch. We can then apply the dense pixel-wise advantage that combines *image-level* reward $R$ and *pixel-level* reward $R_P$:

$$\hat{A}_t^i(h, w) = \frac{R_D(\boldsymbol{x}_0^i, \boldsymbol{c}, h, w) - \text{mean}(\{R_D(\boldsymbol{x}_0^i, \boldsymbol{c}, h, w)\}_{i=1}^G)}{\text{std}(\{R_D(\boldsymbol{x}_0^i, \boldsymbol{c}, h, w)\}_{i=1}^G)}.$$
(5)

where $R_D(\boldsymbol{x}_0^i, \boldsymbol{c}, h, w) = R(\boldsymbol{x}_0^i, \boldsymbol{c}) \cdot (1 - R_P(\boldsymbol{x}_0^i, \boldsymbol{c}, h, w))$ is the dense reward function.

The DenseFlow-GRPO objective is defined as:

$$\mathcal{J}_{\text{Dense}}(\phi) = \frac{1}{GTHW} \sum_{i,t,h,w} \left( \min\left(s_t^i(\phi, h, w)\hat{A}_t^i(h, w), \, \text{clip}\left(s_t^i(\phi, h, w), 1-\varepsilon, 1+\varepsilon\right)\hat{A}_t^i(h, w)\right) \right),$$
(6)

where we omit the KL regularization term for brevity. Note that in Eq. (4), $s_t^i(\phi, h, w)$ is numerically equal to $r_t^i(\phi)$ but allows the pixel-wise advantage to backpropagate to the local image regions through $p_\phi(\cdot)_{h,w}$. This is similar to the GSPO-token (Zheng et al., 2025) formulation, and we find it more stable than directly computing the pixel-wise likelihood ratio.

# 5 EXPERIMENTS

## 5.1 EXPERIMENTAL SETUP

**Datasets.** We train and evaluate ImageDoctor on the *RichHF-18K* (Liang et al., 2024) dataset. RichHF-18K is a subset of Pick-a-Pic (Kirstain et al., 2023), consisting of 16K training samples,

Table 1: Performance comparison of score prediction on RichHF-18K.

| Method | Plausibility | | Aesthetics | | Semantic Alignment | | Overall | | Average | |
|---|---|---|---|---|---|---|---|---|---|---|
| | PLCC ↑ | SRCC ↑ | PLCC ↑ | SRCC ↑ | PLCC ↑ | SRCC ↑ | PLCC ↑ | SRCC ↑ | PLCC ↑ | SRCC ↑ |
| ResNet-50 (He et al., 2016) | 0.495 | 0.487 | 0.370 | 0.363 | 0.108 | 0.119 | 0.337 | 0.308 | 0.328 | 0.319 |
| CLIP (Radford et al., 2021) | 0.390 | 0.378 | 0.357 | 0.360 | 0.398 | 0.390 | 0.353 | 0.352 | 0.374 | 0.370 |
| PickScore (Kirstain et al., 2023) | 0.010 | 0.028 | 0.131 | 0.140 | 0.346 | 0.340 | 0.202 | 0.226 | 0.172 | 0.183 |
| RichHF (Liang et al., 2024) | 0.693 | 0.681 | 0.600 | 0.589 | 0.474 | 0.496 | 0.580 | 0.562 | 0.586 | 0.582 |
| **ImageDoctor (Ours)** | **0.727** | **0.711** | **0.681** | **0.662** | **0.808** | **0.799** | **0.745** | **0.725** | **0.741** | **0.724** |

Table 2: Performance comparison of heatmap prediction on RichHF-18K.

| Method | Artifact | | | | | Misalignment | | | | |
|---|---|---|---|---|---|---|---|---|---|---|
| | All data | $GT = 0$ | $GT > 0$ | | | All data | $GT = 0$ | $GT > 0$ | | |
| | MSE ↓ | MSE ↓ | CC ↑ | KLD ↓ | SIM ↑ | MSE ↓ | MSE ↓ | CC ↑ | KLD ↓ | SIM ↑ |
| ResNet-50 (He et al., 2016) | 0.00996 | 0.00093 | 0.506 | 1.669 | 0.338 | - | - | - | - | - |
| CLIP Gradient (Simonyan et al., 2013) | - | - | - | - | - | 0.00817 | 0.00551 | 0.015 | 3.844 | 0.041 |
| RichHF (Liang et al., 2024) | 0.00920 | 0.00095 | 0.556 | 1.652 | 0.409 | 0.00304 | 0.00006 | 0.212 | 2.933 | 0.106 |
| **ImageDoctor (Ours)** | **0.00891** | **0.00076** | **0.571** | **1.477** | **0.412** | **0.00299** | **0.00003** | **0.225** | **2.863** | **0.108** |

1K validation samples, and 1K test samples. For each text-image pair, it provides two heatmaps and four fine-grained scores annotated by a total of 27 annotators. To further assess the generalizability of ImageDoctor, we also test it on the *GenAI-Bench* (Li et al., 2024) and *TIFA* (Hu et al., 2023).

**Evaluation Metrics.** We follow the official evaluation protocols of the datasets. For score prediction, we employ Pearson Linear Correlation Coefficient (PLCC) and Spearman Rank Correlation Coefficient (SRCC), which measure how well the predicted scores correlate with human annotations. For heatmap prediction, we report the Mean Squared Error (MSE) between predictions and ground truth. Additionally, we adopt standard heatmap metrics (Liang et al., 2024) including KL Divergence (KLD), Similarity (SIM), and Correlation Coefficient (CC), providing a comprehensive assessment of spatial prediction quality.

**Implementation Details** All experiments are conducted on four AMD MI250 GPUs. We adopt Qwen2.5-VL-3B (Bai et al., 2025) as the MLLM backbone. Training is performed for 5 epochs in cold start stage 1 and 3 epochs in stage 2. We train for 400 steps for RFT. Learning rates are set to $1 \times 10^{-5}$ for cold start and $1 \times 10^{-6}$ for RFT. Training images are resized to $512 \times 512$.

## 5.2 MAIN RESULTS

**Results on RichHF-18K.** Table 1 shows the score prediction results across four dimensions on RichHF-18K. We compare with ResNet-50 (He et al., 2016), CLIP (Radford et al., 2021) models fine-tuned on the RichHF-18K dataset, as well as the off-the-shelf PickScore (Kirstain et al., 2023) model. In addition, we compare with the RichHF model (Liang et al., 2024), which is trained on RichHF-18k and is able to generate both score and heatmap predictions. ImageDoctor achieves the best performance across all dimensions, substantially improving semantic alignment (PLCC: 0.808 vs. 0.474) and raising the average PLCC from 0.586 to 0.741 compared to the previous best method RichHF, demonstrating much stronger correlation with human judgment. These gains also extend to heatmap prediction results in Table 2, where ImageDoctor achieves the best performance, highlighting its ability to precisely localize flaws in generated images.

**Results on GenAI-Bench and TIFA.** Table 3 presents the results on the GenAI-Bench and TIFA datasets. To assess the generalizability of ImageDoctor, we evaluate the model trained solely on RichHF-18K without any fine-tuning on these two benchmarks. Despite differences in image sources and the inherent subjectivity of annotators, ImageDoctor consistently outperforms previous human preference models, achieving higher correlations with human annotations.

**Heatmap Visualization.** In Fig. 3, we present qualitative examples of heatmap predictions generated by ImageDoctor. For the misalignment heatmaps (Fig. 3 (a)), our model accurately localizes objects that fail to correspond to the prompt while producing fewer false positives. For the artifact heatmaps (Fig. 3 (b)), ImageDoctor effectively highlights all the regions containing artifacts, demonstrating precise spatial grounding of visual flaws. More examples are provided in Section E.4.

Table 3: Quantitative comparison on the GenAI-Bench and TIFA datasets.

| Method | GenAI-Bench PLCC | TIFA PLCC | TIFA SRCC | RichHF PLCC | RichHF SRCC |
|---|---|---|---|---|---|
| CLIPScore | 0.164 | 0.309 | 0.300 | 0.302 | 0.057 |
| ImageReward | 0.350 | 0.633 | 0.621 | 0.329 | 0.274 |
| PickScore | 0.354 | 0.413 | 0.392 | 0.346 | 0.340 |
| HPSv2 | 0.139 | 0.380 | 0.365 | 0.258 | 0.187 |
| VQAScore | 0.499 | 0.659 | 0.695 | 0.409 | 0.483 |
| EvalMuse | 0.498 | 0.712 | 0.749 | 0.549 | 0.518 |
| HPSv3 | 0.139 | 0.485 | 0.484 | 0.205 | 0.184 |
| **ImageDoctor** | **0.514** | **0.740** | **0.764** | **0.808** | **0.799** |

Table 4: **Ablation study** of the proposed modules.

| Settings | Average PLCC ↑ | Average SRCC ↑ | Artifact CC ↑ | Artifact KLD ↓ | Misalignment CC ↑ | Misalignment KLD ↓ |
|---|---|---|---|---|---|---|
| **Cold Start Stage 1** | 0.660 | 0.656 | - | - | - | - |
| + Heatmap | 0.655 | 0.650 | 0.532 | 1.597 | 0.165 | 3.031 |
| + Heatmap w/o task token | 0.653 | 0.645 | 0.508 | 1.728 | 0.123 | 3.231 |
| **Cold Start Stage 2** | 0.720 | 0.707 | 0.558 | 1.533 | 0.224 | 2.982 |
| w/o *"look"* | 0.714 | 0.705 | 0.534 | 1.599 | 0.160 | 3.131 |
| w/o *"think"* | 0.708 | 0.698 | 0.542 | 1.592 | 0.190 | 3.038 |
| **Reinforcement Finetuning** | 0.741 | 0.724 | 0.571 | 1.477 | 0.225 | 2.863 |
| w/o grounding reward | 0.734 | 0.718 | 0.566 | 1.507 | 0.225 | 2.865 |

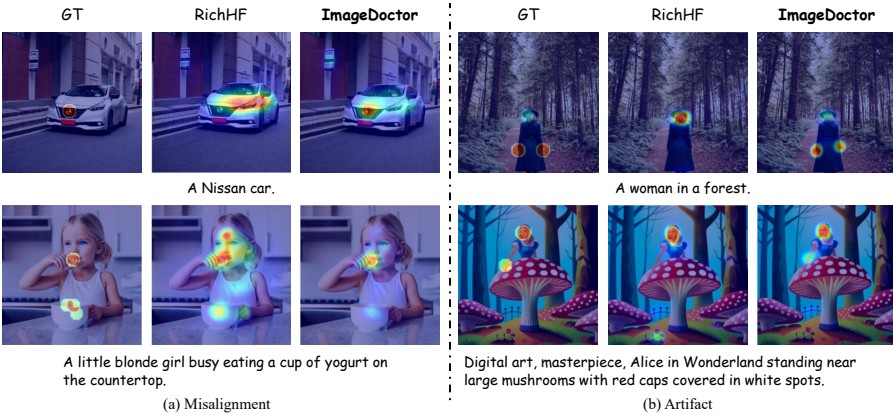

(a) Misalignment      (b) Artifact

Figure 3: Visualization of misalignment and artifact heatmaps.

## 5.3 ABLATION STUDY

We conduct ablation experiments on the RichHF-18K dataset to analyze the contribution of each proposed module in ImageDoctor. The results are shown in Table 4.

**Task token for heatmap decoder.** We introduce the special task tokens in the MLLM backbone to guide the heatmap prediction in the heatmap decoder. To demonstrate its effectiveness, we finetune the ImageDoctor model after cold start stage 1 on both score and heatmap prediction tasks with and without the task tokens. As shown in Table 4 (rows 2 and 3), removing the task tokens results in notable decrease of heatmap prediction performance, *e.g.*, CC drops by 0.024 and 0.042 for artifact and misalignment, respectively.

**Effect of *"look"* and *"think"*.** We propose a *"look-think-prediction"* paradigm for T2I evaluation, where the model first localizes potential flawed regions by predicting bounding boxes (*"look"*), and then analyzes and reasons about these flaws (*"think"*) before making predictions. We conduct ablation studies assessing their contribution in Table 4 (rows 5 and 6). Removing either component leads to a performance drop. In particular, *"think"* plays a more critical role in score accuracy, with PLCC decreasing from 0.720 to 0.708 without *"think"*, while *"look"* provides stronger benefits for heatmap prediction, where misalignment CC falls from 0.224 to 0.160 without *"look"*. These results highlight the complementary roles of *"look"* and *"think"*: the former enhances spatial localization of flaws, while the latter strengthens semantic reasoning for accurate evaluation.

**Effect of grounding reward.** We introduce a grounding reward in reinforcement finetuning to encourage the model to accurately localize flawed regions. As shown in Table 4 (row 8), removing the grounding reward leads to a decline in score prediction performance. Moreover, it also causes a notable drop in artifact heatmap quality. These results hightlight the importance of grounding reward.

## 6 APPLICATION IN DOWNSTREAM TASKS

To further demonstrate the effectiveness of ImageDoctor's rich feedback, we explore its application in downstream tasks, specifically as a verifier in test-time scaling and a reward function for reinforcement learning of T2I model.

## 6.1 IMAGEDOCTOR AS A VERIFIER FOR TEST-TIME SCALING

Recent works have explored test-time scaling (Ma et al., 2025a; Guo et al., 2025) for improving diffusion model performance by generating multiple samples during inference and searching for the best candidate. This approach requires a verifier to distinguish subtle differences among generated images and reliably select the best candidate. We test ImageDoctor as an image verifier, where we sample 16 images for a given prompt and select the best candidates leveraging the four-dimensional scores. The images are generated at a resolution of $1024 \times 1024$ using the Flux-dev (Labs, 2024) model. Visualization results in Fig. 4 show that ImageDoctor reliably selects images that better align with the prompt, often preferring those with more realistic and coherent details.

## 6.2 IMAGEDOCTOR AS A REWARD FUNCTION

**Setup.** We use Stable Diffusion 3.5-medium as the base model and demonstrate the results of using ImageDoctor as a reward function in Flow-GRPO as well as the proposed DenseFlow-GRPO (Sec. 4). We train the models for 1,300 iterations on the Pick-a-Pic prompts (Kirstain et al., 2023). We evaluate the base and finetuned model performance on DrawBench (Saharia et al., 2022) using ImageReward (Xu et al., 2023), CLIPScore (Hessel et al., 2021) and UnifiedReward (Wang et al., 2025b) as the metrics.

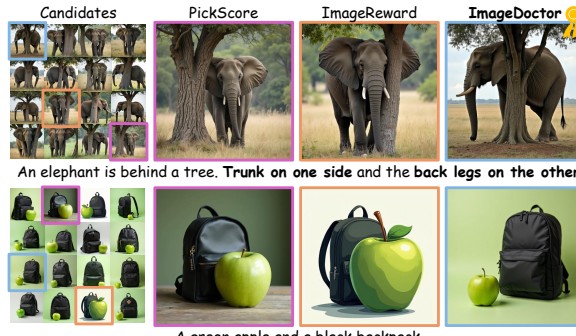

Figure 4: Qualitative comparison on selected images by different verifiers in test-time scaling. ImageDoctor picks the images that faithfully reflect the text prompt (*top*) and preserve realistic object scale (*bottom*).

**Results with Flow-GRPO.** Flow-GRPO adopts score-only reward functions for training T2I model. We use ImageDoctor predicted scores as the reward function, and compare the performance of using PickScore (Kirstain et al., 2023) and RichHF (Liang et al., 2024). As shown in Table 5, using ImageDoctor as the reward function consistently offers the highest gain across all evaluation metrics thanks to its strong ability in predicting accurate image evaluation scores.

Table 5: Performance on human preference scores when ImageDoctor serves as a reward function.

| Reward | ImageReward | CLIPScore | UnifiedReward |
|---|---|---|---|
| Base | 0.818 | 0.951 | 2.903 |
| **Flow-GRPO** | | | |
| PickScore | 1.002 | 0.941 | 2.940 |
| RichHF | 0.879 | 0.944 | 2.921 |
| **ImageDoctor** | 1.029 | 0.956 | 2.960 |
| **DenseFlow-GRPO** | | | |
| **ImageDoctor** | 1.100 | 0.969 | 3.000 |

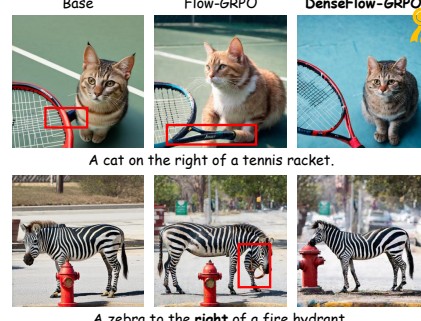

Figure 5: Flow-GRPO vs. DenseFlow-GRPO. The artifacts are boxed.

**Results with DenseFlow-GRPO.** ImageDoctor is capable of generating spatial heatmaps and scalar scores, making it well-suited for pixel-level feedback. To leverage this property, we introduce DenseFlow-GRPO, which incorporates heatmap-guided dense rewards for more fine-grained optimization. As shown in Table 5, by combining scalar scores with heatmaps, DenseFlow-GRPO achieves the best overall results and outperforms the Flow-GRPO variant with ImageDoctor score prediction as the reward function. These findings demonstrate that ImageDoctor's dense, multi-aspect feedback provides fine-grained supervision and leads to consistently stronger alignment with human preference. We provide a visual comparison of Flow-GRPO and DenseFlow-GRPO in Fig. 5. Flow-GRPO adopts an image-level formulation that is often insufficient for removing localized arti-

facts, as the reward signal provides a sparse global score for the entire image, while DenseGRPO's heatmap-based dense reward design can target and refine local details to eliminate such flaws.

# 7 CONCLUSION

In this work, we introduce ImageDoctor, a unified evaluation framework for text-to-image generation that produces both multi-aspect scores and spatially grounded heatmaps. To enhance the evaluation accuracy and interpretability, we propose a *"look-think-predict"* paradigm, which localizes flaws, analyzes them, and delivers a final judgment. Furthermore, we propose DenseFlow-GRPO that utilizes the dense rewards generated by ImageDoctor for finetuning T2I model. Extensive experiments demonstrate its versatility—serving as a metric, verifier, and reward function—showing that ImageDoctor provides robust, interpretable, and human-aligned feedback for generated images.

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

# APPENDIX

## A  DETAILS OF HEATMAP DECODER

To equip the MLLM with the ability to generate accurate heatmaps, we design a lightweight heatmap decoder. The decoder takes image features extracted by the visual encoder, along with a learned heatmap token and a task token that specifies the type of heatmap to be produced. First, the special tokens (task and heatmap) serve as queries to attend over image embedding, after which a multi-layer perceptron (MLP) updates all special tokens. Next, the updated special tokens are used as keys and values to and the image embeddings act as queries to refine the image features. The updated image embeddings are then passed through a series of convolution and deconvolution layers to upscale to the original image size. Finally, before applying the sigmoid activation, we introduce an additional token-to-image attention: the updated image embeddings attend once more to the special token embeddings. The attended heatmap token features are projected through another MLPs, and their outputs are combined with the upsampled image embeddings via a spatial point-wise product. This design strengthens the role of the task tokens in guiding the final heatmap prediction, ensuring that both semantic reasoning and localized visual evidence contribute to the spatial diagnosis. ⇒ Back to main paper Section 3.1.

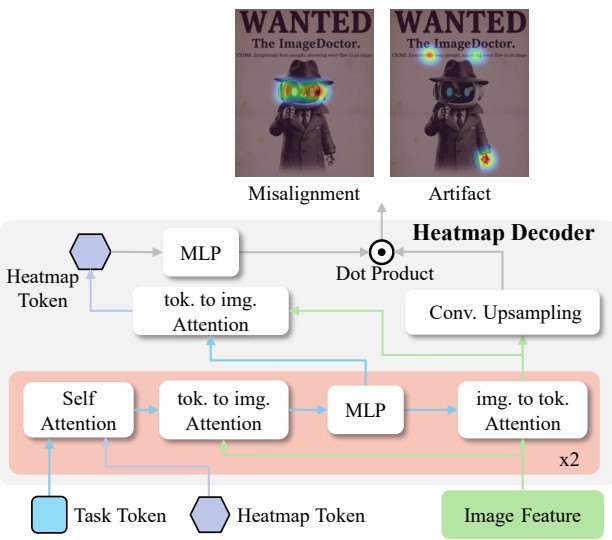

Figure 6: The architecture of heatmap decoder.

## B  GROUNDING IMAGE REASONING GENERATION

For reasoning path generation in Stage 2 of the cold start phase, we employ Gemini-flash with carefully designed prompts to produce detailed reasoning that bridges the image and its human annotations. Since advanced VLMs possess strong visual grounding capabilities, we enrich the input beyond the original image–prompt pair by also providing human-annotated heatmaps indicating the locations of artifacts and misalignments. To preserve the fidelity of the original image, we highlight these flawed regions using circled outlines derived from the heatmaps, rather than directly overlaying them, which enables the VLM to more accurately localize unsatisfactory regions. By combining these localized observations with human-provided scores, Gemini 2.5 Flash generates reasoning that is more precise and relevant to the evaluated image. This high-quality reasoning chain is then used to fine-tune Qwen2.5-VL, equipping it with structured and grounded evaluation capabilities. ⇒ Back to main paper Section 3.2.2.

---

**Prompt for Gemini 2.5 Flash 2.5 for Reasoning Data Generation**

```
You are a multi-modal AI assistant tasked with generating
a reasoning process for a human evaluation of a generated
image.  I am providing three images in a specific order:
1.  The First Image provided is the 'Original Image':  This
is the image generated by a text-to-image model based on
the input prompt "PROMPT".  This Original Image is the SOLE
subject of your evaluation.
2.  The Second Image provided is the 'Artifact Heatmap
Image':  This image is a visual aid.  It is the Original
Image with an overlay that ONLY serves to visually pinpoint
the artifact locations.  Its ONLY purpose is to help you
locate the specified coordinates on the Original Image and
then describe the visual characteristics of the artifact
locations *on the Original Image*.
3.  The Third Image provided is the 'Misalignment Heatmap
Image':  This image is a visual aid.  It is the Original
Image with an overlay that ONLY serves to visually pinpoint
the misalignment locations.  Its ONLY purpose is to help you
locate the specified coordinates on the Original Image and
then describe the visual characteristics of the misalignment
locations *on the Original Image*.
Below, you will find the human evaluation data of the
Original Image for several dimensions, including scores,
keyword alignment status..  Your goal is to analyze the
Original Image and articulate a plausible step-by-step
reasoning that would lead to the given scores, speaking from
the perspective of the evaluator.
Human Evaluation Results:
* Semantic Alignment:  How well the image content corresponds
to the original caption.
* Score:  MISALIGNMENT SCORE
* Aesthetics:  Assessment of composition, color usage, and
overall artistic quality.
* Score:  AESTHETIC SCORE
* Plausibility:  Realism and visual fidelity of the Original
Image, including distortions or unnatural details.
* Score:  ARTIFACT SCORE
* Overall Impression:  General subjective assessment of the
image's quality.
* Score:  OVERALL SCORE
Your Task:  For each of the four evaluation dimensions
(Semantic Alignment, Aesthetics, Plausibility, and Overall
Impression), please provide a paragraph explaining your
reasoning for the score, as if you were the original human
evaluator assessing the Original Image.
* Refer to specific visual elements of the Original Image
that support your reasoning.
* For the "Plausibility" or "Semantic Alignment" dimension:
Refer to the Second Image (Artifact Heatmap Image) or Third
Image (Misalignment Heatmap Image) to visually locate these
coordinates on the Original Image, specifically connect your
reasoning to these coordinates with the help of provided
Artifact or Misalignment Heatmap Image.  Then, describe the
visual nature of the artifact or misalignment locations as it
appears *on the Original Image*.
* For other dimensions, if relevant, explain any potential
```

```
reasons for a score less than perfect by examining the image.
Consider all provided human evaluation results, including
any labels or listed misaligned keywords, in your reasoning.
Output Format for Each Dimension:  Conclude each paragraph
with a sentence in the following strict format:  "Therefore,
I give it a score of X.XX."
Important Instructions:
- Do not mention the artifact or misalignment heatmap images.
- Use the coordinates to focus your visual inspection, but
describe only what is visible in the Original Image.
- Be concise and direct in each evaluation.
- Do not include specific coordinates in your reasoning, just
refer to the visual characteristics at those locations.
Please now provide the reasoning for each dimension,
focusing your analysis and descriptions on the First Image
(the 'Original Image' of the "PROMPT") , using the Second
Image (the 'Artifact Heatmap Image') and Third Image (the
'Misalignment Heatmap Image') strictly as a visual guide to
locate artifact and misalignment locations on the Original
Image.  Do not mention the Artifact or Misalignment Heatmap
Images in your reasoning, only use them to locate coordinates
visually on the Original Image.  As if human evaluation
results and heatmap are not available, you only have the
Original Image to evaluate.
Be precise, concise, and strictly refer to the Original
Image in all visual descriptions.  For each dimension, use
two sentences:  one for the reasoning and one for the score
conclusion.
Begin Evaluation:
```

## C  RFT FORMULATION

Given a pair of input $(I, P)$, ImageDoctor as the policy model $\pi_\theta$, generates a group of $N$ candidate responses $\{o^1, \ldots, o^N\}$. For each response $o^i$, we compute a reward score $\mathcal{R}^i$ using a combination of reward functions. The rewards are normalized within the group to compute the group-normalized advantage:

$$A^i \;=\; \frac{\mathcal{R}^i - \text{mean}(\{\mathcal{R}^i\}_{i=1}^N)}{\text{std}(\{\mathcal{R}^i\}_{i=1}^N)}. \tag{7}$$

We update the policy model $\pi_\theta$ by maximizing the GRPO objective function:

$$\mathcal{J}_{\text{RFT}}(\theta) = \frac{1}{N} \sum_{i=1}^N \left[ \min\left( w^i A^i, \; \text{clip}(w^i, 1 - \epsilon, 1 + \epsilon) A^i \right) - \beta D_{\text{KL}}(\pi_\theta \parallel \pi_{\text{ref}}) \right], \tag{8}$$

where $w^i = \frac{\pi_\theta(o^i | I, P)}{\pi_{\text{old}}(o^i | I, P)}$ denotes the likelihood ratio between $\pi_\theta$ and the old policy model $\pi_{\text{old}}$. The clipping threshold $\epsilon$ regulates the extent to which the policy model may update in each step to stabilize training. $\beta$ controls the KL divergence regularization term that constrains $\pi_\theta$ to remain close to the reference model $\pi_{\text{ref}}$, which is the model at the start of reinforcement learning. $\Rightarrow$ Back to main paper Section 3.2.2.

## D  GROUNDING REWARD DETAILS

To fulfill the three criteria mentioned in the main paper, we design three sub-rewards. First, we compute the average intensity within each bounding box and then take the mean across all boxes. This encourages the model to identify compact regions that align with the highlighted areas, yielding

higher rewards when bounding boxes accurately capture potential flaws. Second, we measure the Intersection over Union (IoU) between each pair of bounding boxes and apply a penalty for large overlaps, which discourages redundant box predictions and promotes compactness in number. Finally, we compute the ratio between the area covered by the union of all bounding boxes and the total intensity of the heatmap, ensuring that the highlighted regions are fully covered. When the heatmap is blank and no bounding boxes are predicted, we assign a reward of 1. Conversely, if a heatmap contains highlighted regions but no bounding boxes are predicted, or if bounding boxes are predicted on a blank heatmap, we assign a reward of 0. ⇒ Back to main paper Section 3.2.2.

# E  EXPERIMENTAL DETAILS

## E.1  DATASETS

**RichHF-18K:** is a subset of the Pick-a-Pic, consisting of 16K training samples, 1K validation samples, and 1K test samples. For each text–image pair, two heatmaps and four fine-grained scores are annotated by a total of 27 annotators.

**GenAI-Bench**: It contains 1,600 prompts designed to cover essential visuo-linguistic compositional reasoning skills, with prompts sourced from professional graphic designers experienced in T2I systems. More than 15,000 human ratings are collected across ten different T2I models, ensuring both diversity and difficulty.

**TIFA**: The test set includes 800 generated images based on 160 text inputs from TIFA v1.0. These images are produced by five generative models and annotated by two independent annotators, providing additional benchmarks for evaluating generalization.

## E.2  EXTENDED QUANTITATIVE RESULTS

In Table 6, we provide additional quantitative results on the RichHF-18K dataset. Compared with the main paper, we include the self-evaluated baseline RichHF and ImageDoctor with the fast inference mode, with additional heatmap metric Normalized Scanpath Saliency (NSS) and AUC-Judd. The fast inference mode is motivated by scenarios where only quantitative scores are needed for efficiency, such as when serving as a reward function. To enable this, we append the input prompt with a fixed reasoning template—`<think> </think> <answer>`—as if the reasoning chain were already completed, allowing ImageDoctor to directly output scores and heatmaps without explicit reasoning chains. As shown in Table 6, this variant incurs only a minor performance drop compared to the full model, while maintaining high accuracy with significantly improved speed. For this reason, we adopt this faster variant in Flow-GRPO and DenseFlow-GRPO, where efficiency in advantage computation is critical.

## E.3  QUALITATIVE RESULTS

Fig. 7 presents ImageDoctor's responses given an image–prompt pair. We observe that ImageDoctor first localizes potential flaw regions, where its reasoning and heatmap predictions closely align. For example, it correctly identifies misaligned keywords such as *drinking lemonade* and *making a lot of phone calls*. In addition, it detects artifacts appearing in the image, including unnatural glass shapes, distorted hands, unrealistic liquid in the glass, and the phone. Finally, the heatmaps accurately depict the misaligned and implausible areas, highlighting ImageDoctor's strong localization and reasoning capabilities and alignment with human preferences. More qualitative results are shown in Figs. 10 to 12.

## E.4  ADDITIONAL HEATMAP VISUALIZATION

As shown in Fig. 8, we provide additional heatmaps for qualitative comparison. ⇒ Back to main paper Section 5.3.

Table 6: Score prediction and heatmap prediction results on RichHF-18K. ↓ indicates lower is better and ↑ indicates higher is better. $GT = 0$ refers to empty ground truth heatmap. $GT > 0$ refers to heatmaps with artifact of misalignment. There are total 69 and 144 out of 955 test samples are empty for artifact and misalignment heatmaps. RichHF: results reported in the paper. RichHF[1]: self-evaluated results from the official checkpoint.

(a) Performance comparison of score prediction.

| | Plausibility | | Aesthetics | | Semantic Alignment | | Overall | | Average | |
|---|---|---|---|---|---|---|---|---|---|---|
| | PLCC ↑ | SRCC ↑ | PLCC ↑ | SRCC ↑ | PLCC ↑ | SRCC ↑ | PLCC ↑ | SRCC ↑ | PLCC ↑ | SRCC ↑ |
| ResNet-50 (He et al., 2016) | 0.495 | 0.487 | 0.370 | 0.363 | 0.108 | 0.119 | 0.337 | 0.308 | 0.328 | 0.319 |
| CLIP (Radford et al., 2021) | 0.390 | 0.378 | 0.357 | 0.360 | 0.398 | 0.390 | 0.353 | 0.352 | 0.374 | 0.370 |
| PickScore (Kirstain et al., 2023) | 0.010 | 0.028 | 0.131 | 0.140 | 0.346 | 0.340 | 0.202 | 0.226 | 0.172 | 0.183 |
| RichHF (Liang et al., 2024) | 0.693 | 0.681 | 0.600 | 0.589 | 0.474 | 0.496 | 0.580 | 0.562 | 0.586 | 0.582 |
| RichHF[1] (Liang et al., 2024) | 0.704 | 0.694 | 0.636 | 0.618 | 0.563 | 0.602 | 0.648 | 0.634 | 0.638 | 0.637 |
| ImageDoctor-fast | 0.722 | 0.712 | 0.675 | 0.656 | 0.793 | 0.792 | 0.728 | 0.700 | 0.729 | 0.715 |
| **ImageDoctor** | **0.727** | **0.711** | **0.681** | **0.662** | **0.808** | **0.799** | **0.745** | **0.725** | **0.741** | **0.724** |

(b) Performance comparison of artifact heatmap prediction.

| | All data | $GT = 0$ | | | $GT > 0$ | | |
|---|---|---|---|---|---|---|---|
| | MSE ↓ | MSE ↓ | CC ↑ | KLD ↓ | SIM ↑ | NSS ↑ | AUC-Judd ↑ |
| ResNet-50 (He et al., 2016) | 0.00996 | 0.00093 | 0.506 | 1.669 | 0.338 | 2.924 | 0.909 |
| RichHF (Liang et al., 2024) | 0.00920 | 0.00095 | 0.556 | 1.652 | 0.409 | **3.085** | **0.913** |
| RichHF[1] (Liang et al., 2024) | 0.00920 | 0.00080 | 0.545 | 1.568 | 0.375 | 1.808 | 0.893 |
| ImageDoctor-fast | 0.00879 | 0.00091 | 0.569 | 1.483 | 0.405 | 1.877 | 0.903 |
| **ImageDoctor** | **0.00891** | **0.00076** | **0.571** | **1.477** | **0.412** | 1.884 | 0.903 |

(c) Performance comparison of misalignment heatmap prediction.

| | All data | $GT = 0$ | | | $GT > 0$ | | |
|---|---|---|---|---|---|---|---|
| | MSE ↓ | MSE ↓ | CC ↑ | KLD ↓ | SIM ↑ | NSS ↑ | AUC-Judd ↑ |
| CLIP Gradient (Simonyan et al., 2013) | 0.00817 | 0.00551 | 0.015 | 3.844 | 0.041 | 0.143 | 0.643 |
| RichHF (Liang et al., 2024) | 0.00304 | 0.00006 | 0.212 | 2.933 | 0.106 | **1.411** | **0.841** |
| RichHF[1] (Liang et al., 2024) | 0.00300 | 0.00020 | 0.219 | 2.900 | 0.099 | 1.224 | 0.794 |
| ImageDoctor-fast | 0.00310 | **0.00001** | 0.219 | 2.890 | **0.121** | 1.230 | 0.796 |
| **ImageDoctor** | **0.00299** | 0.00003 | **0.225** | **2.863** | 0.108 | 1.257 | 0.801 |

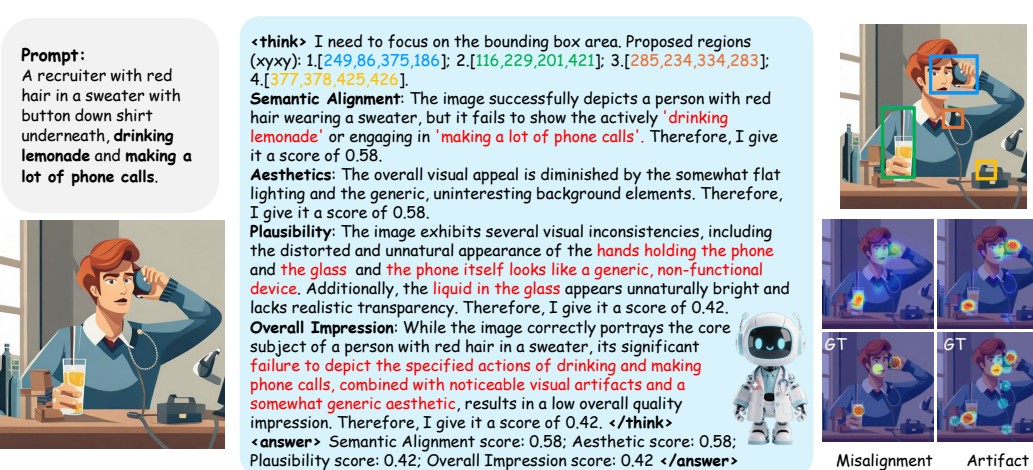

Figure 7: Example of ImageDoctor Predictions.

## E.5 Additional Dense-GRPO Results

In Fig. 8, we provide additional comparison between Flow-GRPO refined with PickScore and DenseFlow-GRPO refined by ImageDoctor. ⇒ Back to main paper Section 5.3.

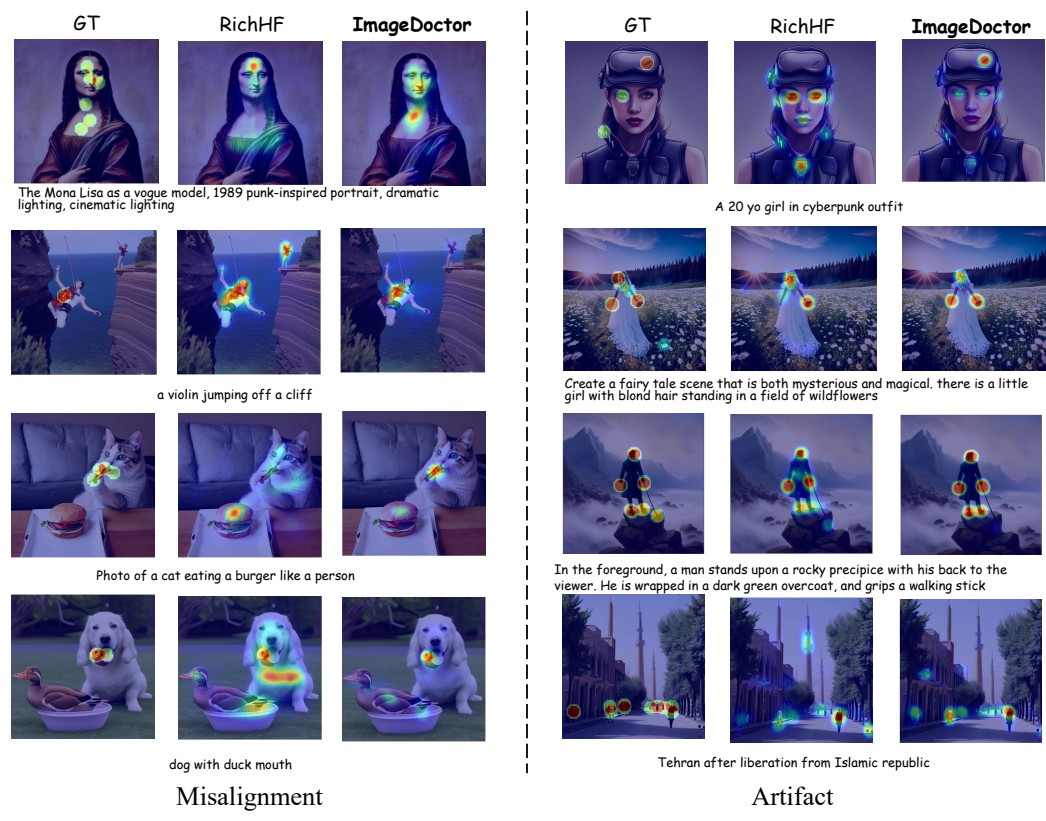

Figure 8: Additional heatmap visualization.

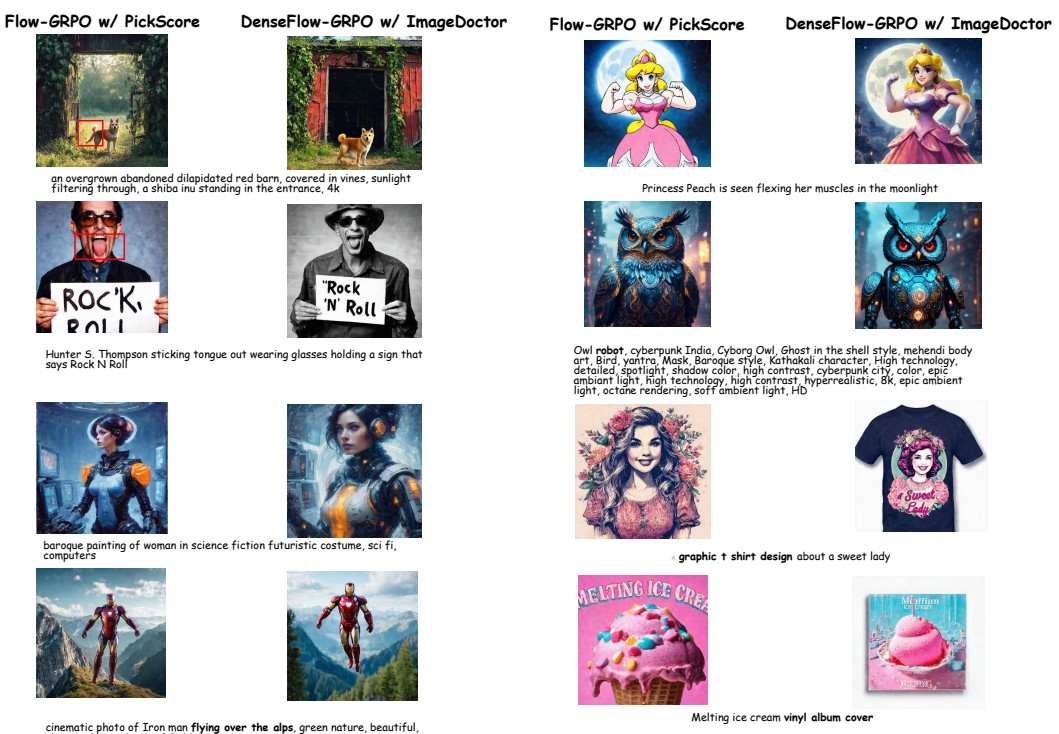

Figure 9: Additional DenseFlow-GRPO visualization.

## F    LIMITATIONS

While ImageDoctor demonstrates strong capability in providing interpretable multi-aspect scoring with spatially grounded feedback, it is important to acknowledge certain limitations that may affect its generalizability and applicability.

**Challenge of large-scale fine-grained annotation.**    Collecting detailed annotations—including multi-aspect scores and heatmaps—is time-consuming and labor-intensive, which limits the volume of available data. The dataset we trained on, i.e. RichHF-18K, is a subset of Pick-a-Pic, which is mostly generated by some old image generation models, *e.g.* Stable Diffusion XL. This constraint in both scale and recency can restrict the full potential of ImageDoctor. Nevertheless, in this work we show that even with limited and somewhat outdated annotations, ImageDoctor can be effectively trained and still provide valuable dense feedback as a verifier, a reward function, and a metric.

**Human preference is subjective.** Quantifying image quality is inherently challenging because different people may perceive the same image in very different ways, making it difficult to establish a universally agreed-upon standard. This subjectivity often results in inconsistent annotations, which can affect dataset quality. It also affects heatmap annotations: for instance, annotators may disagree on the exact regions that constitute misalignment, leading to noisy supervision. Consequently, ImageDoctor achieves lower performance on misalignment heatmaps compared to artifact heatmaps.

**Tradeoff between Interpretability and Efficiency.** While ImageDoctor offers substantially improved interpretability through its reasoning chains and pixel-level diagnostics, this naturally introduces additional inference cost compared to scalar-only evaluators. This reflects an inherent tradeoff between interpretability and efficiency: scalar metrics are compact but cannot explain why an image is deficient, whereas ImageDoctor's structured reasoning enables clear attribution of issues such as aesthetic flaws, semantic misalignment, or implausible regions. To maintain practicality for real-world deployment, we also provide a fast inference mode, in which the model is trained with reasoning chains but prompted to skip them at inference by immediately emitting an end-of-thinking token. This variant preserves comparable accuracy while significantly reducing computational overhead. Together, these two modes offer a flexible spectrum between interpretability and efficiency, allowing users to select the configuration that best fits their application needs.

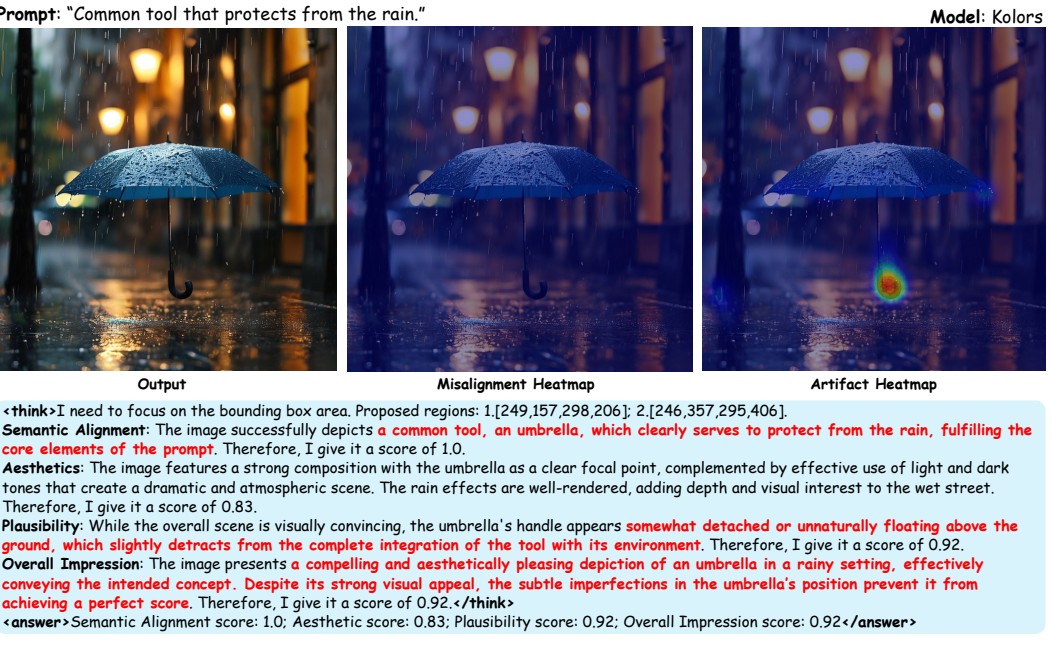

Figure 10: Example of ImageDoctor Predictions with Prompt from WISE (Niu et al., 2025).

**Prompt**: "Traditional food of the Mid-Autumn Festival"                    **Model**: SD3.5-medium

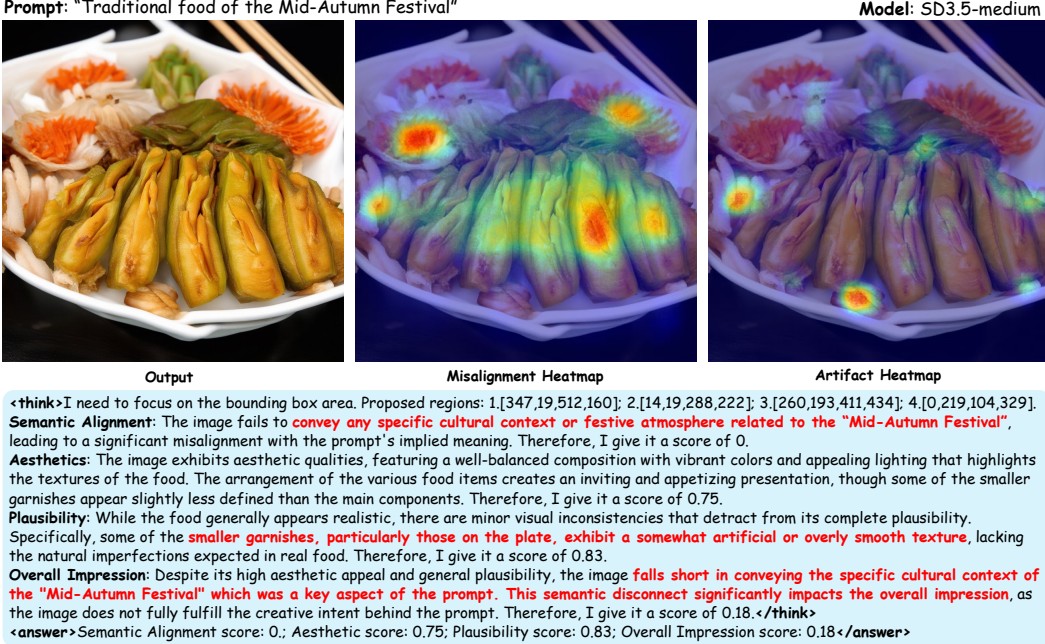

**<think>**I need to focus on the bounding box area. Proposed regions: 1.[347,19,512,160]; 2.[14,19,288,222]; 3.[260,193,411,434]; 4.[0,219,104,329].
**Semantic Alignment**: The image fails to convey any specific cultural context or festive atmosphere related to the "Mid-Autumn Festival", leading to a significant misalignment with the prompt's implied meaning. Therefore, I give it a score of 0.
**Aesthetics**: The image exhibits aesthetic qualities, featuring a well-balanced composition with vibrant colors and appealing lighting that highlights the textures of the food. The arrangement of the various food items creates an inviting and appetizing presentation, though some of the smaller garnishes appear slightly less defined than the main components. Therefore, I give it a score of 0.75.
**Plausibility**: While the food generally appears realistic, there are minor visual inconsistencies that detract from its complete plausibility. Specifically, some of the smaller garnishes, particularly those on the plate, exhibit a somewhat artificial or overly smooth texture, lacking the natural imperfections expected in real food. Therefore, I give it a score of 0.83.
**Overall Impression**: Despite its high aesthetic appeal and general plausibility, the image falls short in conveying the specific cultural context of the "Mid-Autumn Festival" which was a key aspect of the prompt. This semantic disconnect significantly impacts the overall impression, as the image does not fully fulfill the creative intent behind the prompt. Therefore, I give it a score of 0.18.**</think>**
**<answer>**Semantic Alignment score: 0.; Aesthetic score: 0.75; Plausibility score: 0.83; Overall Impression score: 0.18**</answer>**

Figure 11: Example of ImageDoctor Predictions with Prompt from WISE (Niu et al., 2025).

**Prompt**: "The item worn to protect the head while riding a motorcycle."                    **Model**: Flux-schnell

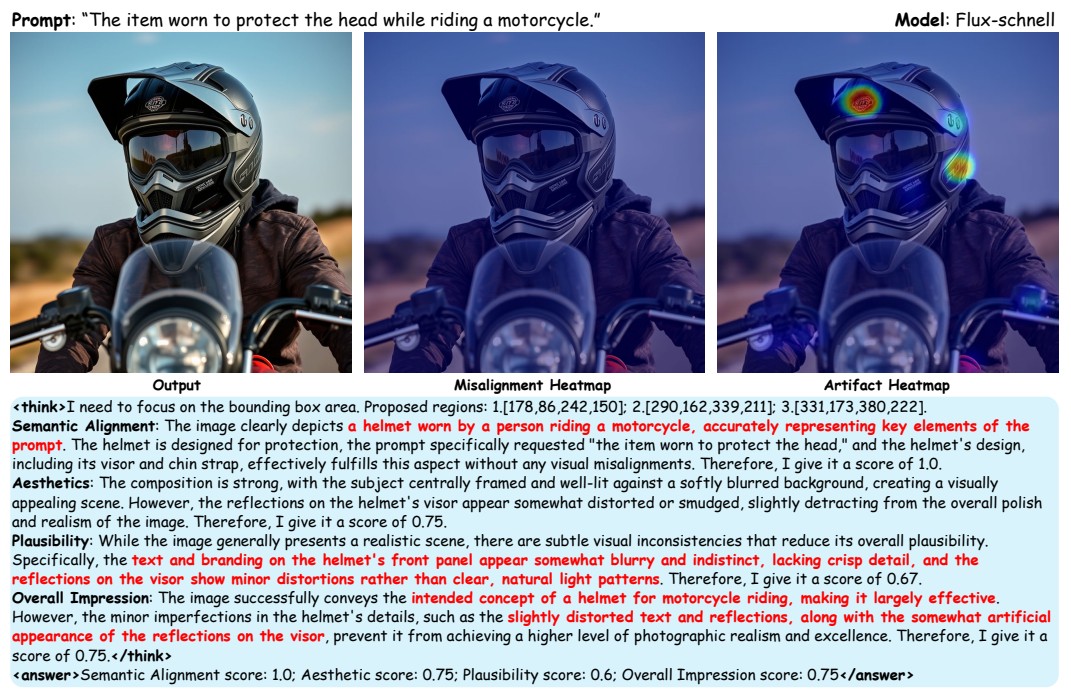

**<think>**I need to focus on the bounding box area. Proposed regions: 1.[178,86,242,150]; 2.[290,162,339,211]; 3.[331,173,380,222].
**Semantic Alignment**: The image clearly depicts a helmet worn by a person riding a motorcycle, accurately representing key elements of the prompt. The helmet is designed for protection, the prompt specifically requested "the item worn to protect the head," and the helmet's design, including its visor and chin strap, effectively fulfills this aspect without any visual misalignments. Therefore, I give it a score of 1.0.
**Aesthetics**: The composition is strong, with the subject centrally framed and well-lit against a softly blurred background, creating a visually appealing scene. However, the reflections on the helmet's visor appear somewhat distorted or smudged, slightly detracting from the overall polish and realism of the image. Therefore, I give it a score of 0.75.
**Plausibility**: While the image generally presents a realistic scene, there are subtle visual inconsistencies that reduce its overall plausibility. Specifically, the text and branding on the helmet's front panel appear somewhat blurry and indistinct, lacking crisp detail, and the reflections on the visor show minor distortions rather than clear, natural light patterns. Therefore, I give it a score of 0.67.
**Overall Impression**: The image successfully conveys the intended concept of a helmet for motorcycle riding, making it largely effective. However, the minor imperfections in the helmet's details, such as the slightly distorted text and reflections, along with the somewhat artificial appearance of the reflections on the visor, prevent it from achieving a higher level of photographic realism and excellence. Therefore, I give it a score of 0.75.**</think>**
**<answer>**Semantic Alignment score: 1.0; Aesthetic score: 0.75; Plausibility score: 0.6; Overall Impression score: 0.75**</answer>**

Figure 12: Example of ImageDoctor Predictions with Prompt from WISE (Niu et al., 2025).

## G PROMPT FOR IMAGEDOCTOR

The prompt to ask ImageDoctor to multi-aspect scores and spatial feedback with reasoning is as following:

---

**Prompt for ImageDoctor for T2I Evaluation**

```
<image>
Given a caption and an image generated based on this caption,
please analyze the provided image in detail.  Evaluate
it on various dimensions including Semantic Alignment
(How well the image content corresponds to the caption),
Aesthetics (composition, color usage, and overall artistic
quality), Plausibility (realism and attention to detail),
and Overall Impression (General subjective assessment of the
image's quality).  For each evaluation dimension, provide
a score between 0-1 and provide a concise rationale for
the score.  Use a chain-of-thought process to detail your
reasoning steps, and enclose all potential important areas
and detailed reasoning within <think> and </think> tags.
The important areas are represented in following format:"
I need to focus on the bounding box area.  Proposed regions
(xyxy):  ..., which is an enumerated list in the exact
format:1.[x1,y1,x2,y2];2.[x1,y1,x2,y2];3.[x1,y1,x2,y2]...
Here, x1,y1 is the top-left corner, and x2,y2 is the
bottom-right corner.  Then, within the <answer> and </answer>
tags, summarize your assessment in the following format:
"Semantic Alignment score:  ...
Aesthetic score:  ...
Plausibility score:  ...
Overall Impression score:  ...
Misalignment Locations:  ...
Artifact Locations:  ..."
No additional text is allowed in the answer section.
Your actual evaluation should be based on the quality of the
provided image.
Your task is provided as follows:
Text Caption:  [PROMPT]
```

---

## H LLM USAGE STATEMENT

We employed Gemini-2.5 Flash for preparing reasoning path generation and ChatGPT5 to refine sentence structure and enhance the readability of the manuscript. In addition, Nano Banana was used to assist in generating illustrative figures for clearer presentation. The LLMs were not involved in research ideation or experimental design. LLM assistance on language editing did not influence the substance of the work.

