# OpenReview forum: "ImageDoctor: Diagnosing Text-to-Image Generation via Grounded Image Reasoning"
_ICLR.cc/2026/Conference — ICLR 2026 Poster_

### Official Review · Reviewer_bUcB · 2025-10-29

**Soundness:** 3
**Presentation:** 2
**Contribution:** 2
**Rating:** 6
**Confidence:** 4

**Summary:**

In this paper,the authors introduces ImageDoctor, a unified,multi-aspect evaluation framework for Text-to Image(T2I) models. Unlike previous methods that provide a single scalar, ImageDoctor assesses image quality across four dimensions: plausibility, semantic alignment, aesthetics, and overall quality.ImageDoctor also provides pixel-level flaw indicators in the form of heatmaps, which highlight misaligned or implausible regions, and can be used as a dense reward for T2I model preference alignment. The model is built on a multi-modal large language models(MLLMs) and adopts a “look-think-predict” paradigm. Training involves a two-phase process: cold start and reinforcement finetuning with Group Relative Policy Optimization(GRPO) using tailored rewards. Furthermore, the paper proposes DenseFlow-GRPO, which utilizes ImageDoctor’s dense, pixel-level heatmaps as a dense reward signal. Experiments demonstrates that ImageDoctor achieves strong alignment with human preference across multiple datasets. Furthermore,when used as a reward model for preference tuning, ImageDoctor achieves an improvement of 10% over scalar-based reward models.

**Strengths:**

1.ImageDoctor breaks through the limitation of a single score by evaluating images from four meaningful dimensions(plausibility, semantic alignment, aesthetics, and overall quality).
2.The ablation study demonstrates that the “look-think-predict” paradigm significantly improves the model’s overall performance.
3.ImageDoctor is further introduced DenseFlow-GRPO, which leads to superior generation quality, especially in refining local details.
4.As a reward model, ImageDoctor improves the quality of T2I model generation by 10% compared to scalar based reward models.
5.ImageDoctor's heatmaps and inference chain can directly locate defect areas, filling the gap of lack of interpretability and spatial localization in T2I evaluation.

**Weaknesses:**

1.ImageDoctor is trained mainly on RichHF-18K,which is limited in size and flaw type. These may affect its generalization ability to state-of-the-art T2I model generated images.
2.ImageDoctor involves a MLLM, a heatmap decoder, a two-phase training pipeline.Complex processes are not replicated and promoted on a large scale.

**Questions:**

1.When there are multiple overlapping flaws in the image, the bounding boxes generated in the "look" stage may have "overlapping coverage". Will this cause the "think" stage to be unable to distinguish between different types of defects?
2.Could the authors provide a detailed explanation through which refining the bounding boxes via reward in the RFT phase leads to more accurate artifact heatmaps and higher score prediction accuracy? Does better "look" translate to better "think"?
3.The MLLM output involves reasoning chains and scores. How critical is the rich textual reasoning generated by the MLLM to the final quality of the heatmaps, beyond just the task token?
4.The paper chooses Qwen2.5-VL as the backbone of MLLM and does not mention the comparison with other mainstream multi-modal models such as Gemini 2.5 Flash and GPT-4o. What is the core reason for choosing Qwen2.5-VL? If replaced with Gemini 2.5 Flash, will ImageDoctor have performance differences in score consistency (PLCC/SRCC) and heatmap quality (MSE/CC)?

---

> ### Author Response · Authors · 2025-11-23
> **Response to Reviewer bUcB (1/2)**
>
> We thank the reviewer for their constructive feedback and provide the following clarifications to address the concerns.
>
> **W1: Generalization to State-of-the-art T2I Model Generated Images.**
>
> **Response:** We appreciate the reviewer’s insightful comment regarding the limited scale and diversity of the RichHF-18K dataset and its potential impact on ImageDoctor’s generalizability. To address this concern, we additionally evaluate ImageDoctor on the WISE benchmark using images generated from more recent advanced T2I models, including Stable Diffusion 3.5-medium, Flux-schnell, and Kolors, which differ substantially from those used in RichHF-18K.
>
> As shown in Fig. 10–12, we are pleased to find that ImageDoctor reliably identifies **violations of physical laws** (e.g., Fig. 10: “the umbrella's handle appears detached or unnaturally floating above the ground”), **commonsense inconsistencies** (e.g., Fig. 11: “the image fails to convey any cultural context or festive atmosphere related to the ‘Mid-Autumn Festival’”), and **diverse artifacts** such as blurriness, abnormal geometry, and other implausible visual structures (e.g., Fig. 12: “text and branding on the helmet's front panel appear somewhat blurry and indistinct, lacking crisp detail, and the reflections on the visor show minor distortions rather than clear, natural light patterns”). These results demonstrate strong generalization and diagnostic precision of ImageDoctor. We believe this robustness stems from both the inherent world knowledge of the MLLM backbone and the grounded image reasoning capabilities introduced by our training strategy. In future work, ImageDoctor can benefit from larger and more diversified training data to further enhance its generalizability.
>
> **W2: Architecture and Training Complexity**
>
> **Response:** We appreciate the reviewer’s concern regarding the perceived complexity of ImageDoctor’s architecture and training pipeline. Although ImageDoctor integrates an MLLM backbone, a lightweight heatmap decoder, and a two-phase training procedure, each component is intentionally designed to be modular, lightweight, and scalable, making the system easy to reproduce and extend:
>
> * **Modular architecture.** The heatmap decoder is a minimal addition (**<1% of total parameters**) and does not alter the backbone MLLM. This modularity allows ImageDoctor to be seamlessly integrated with various open-source MLLMs (e.g., Qwen2.5-VL, InternVL).
>
> * **Standardized training pipeline.** The two-phase pipeline (SFT → RFT) directly follows widely adopted RLHF training paradigms. ImageDoctor does not rely on any custom optimizer or specialized framework—everything is built on standard, publicly available GRPO implementations.
>
> * **Empirically scalable and reproducible.** Despite training on a relatively small dataset (RichHF-18K), ImageDoctor generalizes well to unseen domains (GenAI-Bench, TIFA, WISE). Once similar structured annotations (scores + heatmaps) are available, the same modular pipeline can be scaled to larger datasets without architectural changes.
>
> **Q1: Overlapping Flaws**
>
> **Response:** For efficiency and consistency, we do not generate separate bounding boxes for artifacts and misalignments; instead, we take the union of both heatmaps to produce a single flaw-region bounding box. The purpose of this box is not to classify defect types but to **guide the model’s attention** toward regions that merit closer inspection during the *look* stage.
>
> Even when multiple flaws overlap, this does not hinder the *think* stage. The MLLM uses the bounding box only as a **coarse localization prior**, and the distinction between artifact vs. misalignment is determined through its visual–textual reasoning. Importantly, from the MLLM’s perception standpoint, separating bounding boxes by flaw type offers little additional benefit, because the model ultimately infers defect type from the **image content**, not from externally assigned labels.
>
> Therefore, using a unified bounding box does not prevent the model from distinguishing defect types in the think stage; instead, it ensures that the model focuses its reasoning on the correct regions while preserving efficiency and stability in training.

---

> > ### Author Response · Authors · 2025-11-23
> > **Response to Reviewer bUcB (2/2)**
> >
> > **Q3: How critical is the rich textual reasoning generated by the MLLM to the final quality of the heatmaps?**
> >
> > **Response:** We appreciate the reviewer’s question. In ImageDoctor, the textual reasoning chain is **not** fed directly into the heatmap decoder. The decoder operates on **image features** and **task tokens** (artifact/misalignment). However, the reasoning chain remains important because the task tokens themselves are *generated after* the model performs its “look-think” procedure. During this process, the MLLM attends to the localized flaw regions and organizes its understanding before emitting the final tokens. This internal reasoning leads to **higher-quality task-token embeddings**, which the heatmap decoder uses as conditioning signals.
> >
> > Empirically, Table 4 supports this: removing the “think” stage in Stage 2 leads to drops in accuracy in both score and heatmap prediction. These results show that while the reasoning chain is not directly consumed by the decoder, it significantly **enhances the semantic quality of the task tokens**, thereby improving both score prediction and heatmap localization.
> >
> > **Q4: Choice of MLLM Backbone**
> >
> > **Response:** Thank you for raising this important question about the choice of backbone and the comparison with other mainstream MLLMs. The selection of **Qwen2.5-VL** is primarily driven by two practical and methodological considerations.
> > First, **ImageDoctor requires multi-modal output**—it must generate not only textual reasoning and scalar scores, but also pixel-level heatmaps. Most mainstream MLLMs support multimodal input, but only textual output. Closed-source models such as Gemini 2.5 Flash and GPT-4o **do not allow architectural modification or fine-tuning** to produce spatial heatmaps. In contrast, Qwen2.5-VL is fully **open-source**, making it possible to modify the model structure and train it end-to-end for multi-model output, which is essential for ImageDoctor.
> >
> > Second, we require the MLLM backbone to learn **image evaluation–specific behavior**, which general-purpose MLLMs are not optimized for. Qwen2.5-VL is well known for its strong reasoning and grounding abilities, making it a suitable starting point for our “look–think–predict’’ evaluation pipeline. However, to ensure that ImageDoctor is not tightly coupled to a single architecture, we also conducted additional experiments using **InternVL3-1B**, a different model family with a much smaller size. As shown in Table R7, InternVL-based ImageDoctor still achieves significant improvements over prior work, demonstrating that the proposed training pipeline is backbone-agnostic and generalizes across architectures.
> > Since closed-source MLLMs do not support heatmap generation, we evaluate Gemini 3 Pro and GPT-5 solely on scalar score prediction on RichHF-18K. As shown in Table R8-R9, **these models struggle to produce accurate evaluation scores**, highlighting the challenge of using generic MLLMs—without task-specific finetuning—as evaluation models. This further supports our decision to adopt a trainable open-source backbone.
> >
> > &nbsp;
> > &nbsp;
> >
> > ### Table R7: PLCC Performance Comparison
> >
> > | Method | Plausibility | Aesthetics | Semantic Alignment | Overall | Average |
> > | :--- | :---: | :---: | :---: | :---: | :---: |
> > | RichHF  | 0.693 | 0.600 | 0.474 | 0.580 | 0.586 |
> > | **ImageDoctor (InternVL3)** | 0.701 | 0.659 | 0.711 | 0.683 | 0.689 |
> > | **ImageDoctor (Qwen2.5-VL)** | **0.727** | **0.681** | **0.808** | **0.745** | **0.741** |
> >
> > &nbsp;
> > &nbsp;
> >
> > ### Table R8: PLCC Performance on RichHF-18K against advanced MLLMs
> >
> > | Model | Plausibility | Aesthetics | Semantic Alignment | Overall | Average |
> > | :--- | :---: | :---: | :---: | :---: | :---: |
> > | Gemini-3 Pro| 0.410 | 0.415 | 0.512 | 0.450 | 0.447 |
> > | GPT-5 | 0.206 | 0.359 | 0.471 | 0.413 | 0.363 |
> > | **ImageDoctor** | **0.727** | **0.681** | **0.808** | **0.745** | **0.741** |
> >
> > &nbsp;
> > &nbsp;
> >
> > ### Table R9: SRCC Performance on RichHF-18K against advanced MLLMs
> >
> > | Model | Plausibility | Aesthetics | Semantic Alignment | Overall | Average |
> > | :--- | :---: | :---: | :---: | :---: | :---: |
> > | Gemini-3 Pro| 0.420 | 0.420 | 0.568 | 0.464 | 0.468 |
> > | GPT-5 | 0.186 | 0.363 | 0.552 | 0.435 | 0.384 |
> > | **ImageDoctor** | **0.711** | **0.662** | **0.799** | **0.725** | **0.724** |

---

> > > ### Author Response · Authors · 2025-11-27
> > > **Looking Forward to Your Feedback**
> > >
> > > Dear Reviewer bUcB,
> > >
> > > It seems we have not received your feedback on our response yet. As we enter the final week of the discussion phase, we would like to ensure that we have fully addressed your concerns.
> > >
> > > If there are any remaining questions or points that would benefit from further clarification, we would be grateful for the opportunity to elaborate.
> > >
> > > Thank you again for your time and consideration, and we look forward to your feedback.

---

### Official Review · Reviewer_z1tc · 2025-10-29

**Soundness:** 3
**Presentation:** 3
**Contribution:** 3
**Rating:** 8
**Confidence:** 4

**Summary:**

This paper proposes a novel VLM-based evaluation framework for text-to-image generation, named ImageDoctor. ImageDoctor not only provides multi-dimensional scoring capabilities, such as aesthetics and text-image alignment, but also offers pixel-level localization of flawed regions, enabling it to actively identify areas of misalignment and visual implausibility. Notably, the latter capability introduces a fresh perspective for reward modeling in text-to-image generation. Combined with the authors' proposed DenseFlow-GRPO method, which leverages pixel-level supervision signals for reinforcement learning, the framework effectively enhances the performance of image generation models.

**Strengths:**

- Comprehensive Evaluation Framework: It introduces ImageDoctor, a novel VLM-based framework that provides not only multi-dimensional scoring (e.g., on aesthetics and text-image alignment) but also pixel-level localization of flawed areas, offering a more thorough analysis than mere scoring systems.

- Innovative RL Paradigm for Image Generation: The pixel-level supervision signal from ImageDoctor paves the way for a novel reward model, which, combined with the proposed DenseFlow-GRPO reinforcement learning method, effectively enhances the performance of image generation models.

**Weaknesses:**

- The model's ability to localize flaws is trained on the RF-18K dataset. While effective, this dataset's relatively small scale may constrain the full potential of the Vision-Language Model (VLM), potentially limiting its generalizability and precision in identifying a wider variety of visual anomalies.

- The evaluation is based on GenAI-Bench and TIFA. To more rigorously demonstrate ImageDoctor's advanced capabilities, particularly in pinpointing violations of physical laws and common sense, future work could benefit from testing on newer, more challenging benchmarks such as WISE[1] and ABP[2].

[1] WISE: A World Knowledge-Informed Semantic Evaluation for Text-to-Image Generation

[2] Align Beyond Prompts: Evaluating World Knowledge Alignment in Text-to-Image Generation

**Questions:**

1. Given that ImageDoctor's localization capability is trained on the RF-18K dataset, to what extent can it generalize to detect a diverse range of physically implausible or commonsense-violating anomalies?

2. The authors have successfully engineered a VLM capable of comprehensive image "diagnosis", including flaw localization. Have the authors considered applying this diagnostic capability directly within a unified understanding and generative model? We posit that integrating such a critical feedback mechanism internally could potentially offer more direct and significant improvements to the model's own generative capabilities, perhaps even surpassing the gains achieved by the current approach of using an external reward model for reinforcement learning.

---

> ### Author Response · Authors · 2025-11-23
> **Response to Reviewer z1tc**
>
> We appreciate the reviewer’s valuable insights and take this opportunity to clarify the points raised below.
>
> **W1 & W2 & Q1: Generalization of ImageDoctor**
>
> **Response:** We appreciate the reviewer’s insightful comment regarding the limited scale of the RichHF-18K dataset and its potential impact on ImageDoctor’s generalizability. We agree that the dataset size may constrain the full exploitation of our method. To address this concern, as suggested by the reviewer, we additionally evaluate ImageDoctor on the more challenging **WISE benchmark**, which targets world-knowledge-informed semantic evaluation and differs substantially from RichHF-18K.
>
> As shown in Fig. 10–12, we are pleased to find that ImageDoctor reliably identifies:
>
> * **Violations of physical laws** (Fig. 10): e.g., *“the umbrella's handle appears detached or unnaturally floating above the ground”*.
> * **Commonsense inconsistencies** (Fig. 11): e.g., *“the image fails to convey any cultural context or festive atmosphere related to the ‘Mid-Autumn Festival’”*.
> * **Diverse artifacts** (Fig. 12): such as blurriness, abnormal geometry, and other implausible visual structures, e.g., *“text and branding on the helmet's front panel appear somewhat blurry and indistinct, lacking crisp detail, and the reflections on the visor show minor distortions rather than clear, natural light patterns”*.
>
> These results demonstrate strong generalization and diagnostic precision. We believe this robustness stems from both the inherent world knowledge of the MLLM backbone and the grounded image reasoning capabilities introduced by our training strategy. In future work, ImageDoctor can benefit from larger and more diverse training data to further enhance its generalizability.
>
>
> **Q2: Integration with Unified Model**
>
> **Response:** Thank you for this insightful question. We agree that integrating diagnostic capabilities directly into a unified understanding-generation model is an exciting direction. Our current work positions ImageDoctor as an external reward model because this setup allows easier training and broad applicability across different T2I backbones. Nonetheless, it is possible to train a unified image understanding and generation model that is capable of both image generation and evaluation. Such built-in image diagnostic capability could potentially enable stronger image generation performance and allow multi-step self-refinement image generation. Exploring such integration lies beyond the scope of our current work, but we believe this is a highly promising direction for future work!

---

> > ### Comment · Reviewer_z1tc · 2025-11-24
> >
> > Thank you for your comments. The experiments on the WISE benchmark demonstrate the generalization ability of ImageDoctor. I have no further concerns.

---

> > > ### Author Response · Authors · 2025-11-24
> > >
> > > We are pleased that our response has addressed your concerns. Thank you again for your effort in the review and the discussion!

---

### Official Review · Reviewer_yq1W · 2025-10-31

**Soundness:** 3
**Presentation:** 3
**Contribution:** 3
**Rating:** 6
**Confidence:** 4

**Summary:**

This paper proposes ImageDoctor, a unified framework for Text-to-Image (T2I) evaluation that simultaneously outputs multi-aspect scores and spatially grounded heatmaps, offering richer and more interpretable feedback than traditional single-scalar assessments. The paper also introduces DenseFlow-GRPO, a method for T2I model fine-tuning, with experimental results demonstrating the value of pixel-level feedback in improving evaluation accuracy and eliminating local artifacts.

**Strengths:**

1.  ImageDoctor mitigates the lack of fine-grained diagnostic signals in existing T2I evaluation metrics. Its multi-aspect scores and heatmaps greatly enhance the practicality and interpretability of the feedback.
2.  The proposed DenseFlow-GRPO successfully leverages the dense rewards from ImageDoctor, achieving T2I model optimization superior to that of traditional sparse rewards (Flow-GRPO), and is particularly adept at handling local artifacts.
3.  The experimental results demonstrate a clear advantage performance.

**Weaknesses:**

1.  Does the reward model exhibit "reward hacking" during optimization with DenseFlow-GRPO? How does the paper address this issue, and do the reported experimental results come from trials where reward hacking was successfully mitigated?
2.  Were the same seeds used in the visualization examples for DenseFlow-GRPO and Flow-GRPO to ensure a fair comparison? In the visualization example, a tennis racket still contains artifacts; is this limitation due to the reward model not being strong enough, or is it an inherent issue with the base diffusion model? If it is a problem with the base model, can the effectiveness of the proposed reward model and DenseFlow-GRPO in addressing artifacts be validated using a more powerful base model?
3.  Compared to existing lightweight scalar reward models, what is the specific overhead of ImageDoctor, in terms of inference speed and resource consumption? Please provide a quantitative comparison.
4.  Can you add a discussion about more recent reward models? For example, works like 'Enhancing Reward Models for High-quality Image Generation: Beyond Text-Image Alignment' and 'HPSv3: Towards Wide-Spectrum Human Preference Score'.

**Questions:**

Answer the questions in the weakness.

---

> ### Author Response · Authors · 2025-11-23
> **Response to Reviewer yq1W**
>
> We are grateful for the reviewer’s valuable feedback and have provided further details below to address these points.
>
> **W1: Reward Hacking Concerns in DenseFlow-GRPO**
>
> **Response:** Thank you for raising this important question. We did not observe instances of “reward hacking” during optimization with DenseFlow-GRPO. We believe this is largely because of the design of our reward model, which integrates both image-level scores and pixel-level dense supervision. This combination provides complementary global and local constraints, making it difficult for the model to exploit pathological shortcuts in the reward landscape.
>
> **W2: Visualization Seeds and Artifacts**
>
> **Response:** Thank you for these thoughtful questions. Yes, we used the same random seeds for both DenseFlow-GRPO and Flow-GRPO to ensure a fair comparison. Regarding the remaining artifact on the tennis racket, this issue primarily stems from limitations of the base diffusion model rather than the reward model. As shown in Fig. 5, the base model produces clear artifacts in both the handle and string regions. After applying DenseFlow-GRPO, the fidelity of the generated tennis racket is substantially improved, although subtle imperfections remain due to inherent constraints of the underlying base model. While we believe DenseFlow-GRPO can also be applied to more powerful T2I models, we leave this exploration to future work due to time and hardware resource limitations.
>
> **W3: Computation Overhead of ImageDoctor**
>
> **Response:** Thank you for highlighting the importance of evaluating inference speed and resource consumption. Following your suggestion, we report the average inference time and peak VRAM usage across several representative reward models in Table R5: the lightweight scalar reward models **PickScore** and **ImageReward**, the multi-scalar and heatmap model **RichHF**, and the recent MLLM-based scalar model **HPSv3**. ImageDoctor is indeed more computationally demanding than the lightweight scalar-based reward models because it produces multi-aspect scores, grounded reasoning, and spatial heatmaps. However, **this additional cost yields significantly higher performance (Table 3) and provides better interpretability**.
>
> To address practical deployment concerns, we also explored a **fast inference mode**, where the model is trained with reasoning chains but prompted to skip them during inference to improve inference efficiency. This is achieved by appending an end-of-thinking token `</think>` immediately after `<think>` during inference, forcing the model to produce final scores directly without explicit reasoning. This mode reduces inference time to **one-eighth** of the full version while using **less than half the VRAM** of HPSv3 (Table R5), achieving a similar performance compared to the full version (Table R6). Notably, when compared to RichHF, ImageDoctor-fast delivers **higher accuracy** while requiring **less inference time**, demonstrating a favorable performance–efficiency trade-off for real-world applications.
>
> &nbsp;
> &nbsp;
>
> ### Table R5: Inference Latency and VRAM Consumption
>
> | Evaluators | Latency (s) | Peak VRAM (GB) |
> | :--- | :---: | :---: |
> | PickScore | 0.042 | 3.85 |
> | ImageReward | 0.053 | 1.73 |
> | RichHF | 1.337 | 2.91 |
> | **ImageDoctor** | 3.921 | 7.10 |
> | **ImageDoctor-fast** | 0.564 | 7.10 |
> | HPSv3 | 0.578 | 16.56 |
>
> &nbsp;
> &nbsp;
> &nbsp;
>
> ### Table R6: PLCC Performance Comparison
>
> | Method | Plausibility | Aesthetics | Semantic Alignment | Overall | Average |
> | :--- | :---: | :---: | :---: | :---: | :---: |
> | RichHF  | 0.693 | 0.600 | 0.474 | 0.580 | 0.586 |
> | **ImageDoctor-fast (Qwen2.5-VL)** | 0.722 | 0.675 | 0.793 | 0.728 | 0.729 |
> | **ImageDoctor (Qwen2.5-VL)** | **0.727** | **0.681** | **0.808** | **0.745** | **0.741** |
>
> &nbsp;
> &nbsp;
>
> **W4: Discussion on Recent Reward Models**
>
> **Response:** Thank you for pointing out the importance of discussing recent developments in reward modeling. We have added a discussion on the two recent reward models ICT-HP and HPSv3 in the **related work** section.
>
> >HPSv3 leverages an MLLM backbone to capture rich semantic representations from images and captions and incorporates uncertainty-aware ranking to learn human preferences from paired comparisons. ICT-HP proposes the Image-Contained-Text (ICT) and High-Preference (HP) models by finetuning a CLIP model to evaluate both text-image alignment and the image aesthetics and detail quality. While both demonstrate promising results in predicting image-level scalar scores, they do not provide localized evaluation and offer no interpretability regarding why an image is judged as good or bad.

---

> > ### Author Response · Authors · 2025-11-27
> > **Looking Forward to Your Feedback**
> >
> > Dear Reviewer yq1W,
> >
> > It seems we have not received your feedback on our response yet. As we enter the final week of the discussion phase, we would like to ensure that we have fully addressed your concerns.
> >
> > If there are any remaining questions or points that would benefit from further clarification, we would be grateful for the opportunity to elaborate.
> >
> > Thank you again for your time and consideration, and we look forward to your feedback.

---

### Official Review · Reviewer_LnNs · 2025-11-01

**Soundness:** 3
**Presentation:** 3
**Contribution:** 3
**Rating:** 6
**Confidence:** 2

**Summary:**

This paper presents ImageDoctor, a unified and interpretable evaluation framework for text-to-image generation. ImageDoctor provides multi-dimensional feedback and introduces pixel-level diagnostic heatmaps for grounded and fine-grained evaluation. The model adopts a "look-think-predict" paradigm. Experimental results show that ImageDoctor achieves state-of-the-art correlation with human judgments and improves text-to-image generation quality.

**Strengths:**

The use of heatmap-guided dense rewards for T2I reinforcement learning is  innovative.

Extensive experiments across multiple datasets demonstrate consistent improvement in human alignment metrics.

The paper is well-written and easy to follow.

**Weaknesses:**

ImageDoctor’s reasoning quality may depend heavily on the backbone MLLM (Qwen2.5-VL). The paper lacks analysis on robustness across different model sizes or architectures.


The “look-think-predict” reasoning chain increases inference time compared to scalar evaluators. A discussion of trade-offs between interpretability and efficiency would strengthen the work.


How sensitive is ImageDoctor’s performance to the choice of MLLM backbone? Would smaller or different architectures (e.g., LLaVA, BLIP2) yield similar trends?

**Questions:**

Please refer to the weaknesses.

---

> ### Author Response · Authors · 2025-11-23
> **Response to Reviewer LnNs**
>
> We sincerely thank the reviewer for their insightful comments and wish to offer the following clarifications.
>
> **W1 & W3: Performance of Different Backbone LLM**
>
> **Response:** We thank the reviewers for emphasizing the importance of evaluating ImageDoctor’s robustness across backbone architectures and model sizes. In the main paper, ImageDoctor is implemented with **Qwen2.5-VL-3B**. To further assess architectural and scaling robustness, we additionally conducted experiments using **InternVL3-1B**, a smaller model with a distinct architecture. As shown in Table R1-R4, ImageDoctor built on InternVL3-1B still outperforms the prior state-of-the-art RichHF on the RichHF-18K benchmark, including a notable **+0.237 PLCC** improvement in semantic alignment. These results indicate that **ImageDoctor maintains strong performance across different multimodal LLM backbones**, demonstrating the generalizability of its training strategy and the effectiveness of its grounded image reasoning design.
>
> **W2: Tradeoff between Interpretability and Efficiency**
>
> **Response:** We thank the reviewer for raising this valuable point. We agree that producing a reasoning chain introduces additional tokens, which increases inference latency compared to scalar-only evaluators. This reflects an inherent trade-off between interpretability and efficiency. However, the additional cost brings substantial gains: scalar metrics cannot explain *why* an image is suboptimal, whereas ImageDoctor’s reasoning chain provides structured, interpretable feedback that aligns well with human judgments and clarifies whether issues arise from aesthetics, semantic misalignment, implausible regions, etc.
>
> To mitigate latency concerns, we also explored a **fast inference mode**, where the model is trained with reasoning chains but prompted to skip them during inference to improve inference efficiency. This is achieved by appending an end-of-thinking token `</think>` immediately after `<think>` during inference, forcing the model to produce final scores directly without explicit reasoning. This variant maintains comparable accuracy as shown in Table R1-R4 while significantly reducing the inference time from **3.92s to 0.56s**. This flexible design ensures that **ImageDoctor remains practical for real-world deployment and offers a flexible spectrum between interpretability and efficiency**, allowing users to select the configuration that best fits their application needs.
>
> We have added a dedicated discussion of the interpretability-efficiency trade-off in **Appendix F**.
>
> &nbsp;
> &nbsp;
>
> ### Table R1: PLCC Performance Comparison
>
> | Method | Plausibility | Aesthetics | Semantic Alignment | Overall | Average |
> | :--- | :---: | :---: | :---: | :---: | :---: |
> | RichHF  | 0.693 | 0.600 | 0.474 | 0.580 | 0.586 |
> | **ImageDoctor (InternVL3)** | 0.701 | 0.659 | 0.711 | 0.683 | 0.689 |
> | **ImageDoctor-fast (Qwen2.5-VL)** | 0.722 | 0.675 | 0.793 | 0.728 | 0.729 |
> | **ImageDoctor (Qwen2.5-VL)** | **0.727** | **0.681** | **0.808** | **0.745** | **0.741** |
>
> &nbsp;
> &nbsp;
> &nbsp;
>
> ### Table R2: SRCC Performance Comparison
>
> | Method | Plausibility | Aesthetics | Semantic Alignment | Overall | Average |
> | :--- | :---: | :---: | :---: | :---: | :---: |
> | RichHF  | 0.681 | 0.589 | 0.496 | 0.562 | 0.582 |
> | **ImageDoctor (InternVL3)** | 0.689 | 0.648 | 0.732 | 0.663 | 0.683 |
> | **ImageDoctor-fast (Qwen2.5-VL)** | 0.712 | 0.656 | 0.792 | 0.700 | 0.715 |
> | **ImageDoctor (Qwen2.5-VL)** | **0.711** | **0.662** | **0.799** | **0.725** | **0.724** |
>
>
> &nbsp;
> &nbsp;
>
> ---
>
> &nbsp;
> &nbsp;
>
> ### Table R3: Artifact Heatmap Prediction
> *(MSE ↓, CC ↑, KLD ↓, SIM ↑)*
>
> | Method | All Data (MSE) | GT=0 (MSE) | GT>0 (CC) | GT>0 (KLD) | GT>0 (SIM) |
> | :--- | :---: | :---: | :---: | :---: | :---: |
> | RichHF | 0.00920 | 0.00095 | 0.556 | 1.652 | 0.409 |
> | **ImageDoctor (InternVL3)** | 0.00906 | 0.00079 | 0.556 | 1.533 | 0.409 |
> | **ImageDoctor-fast (Qwen2.5-VL)** | 0.00879 | 0.00091 | 0.569 | 1.483 | 0.405 |
> | **ImageDoctor (Qwen2.5-VL)** | **0.00891** | **0.00076** | **0.571** | **1.477** | **0.412** |
>
> &nbsp;
> &nbsp;
>
> ### Table R4: Misalignment Heatmap Prediction
> *(MSE ↓, CC ↑, KLD ↓, SIM ↑)*
>
> | Method | All Data (MSE) | GT=0 (MSE) | GT>0 (CC) | GT>0 (KLD) | GT>0 (SIM) |
> | :--- | :---: | :---: | :---: | :---: | :---: |
> | RichHF | 0.00304 | 0.00006 | 0.212 | 2.933 | 0.106 | **1.411** | 0.841 |
> | **ImageDoctor (InternVL3)** | 0.00301 | 0.00002 | 0.215 | 2.894 | 0.105 |
> | **ImageDoctor-fast (Qwen2.5-VL)** | 0.00310 | **0.00001** | 0.219 | 2.890 | **0.121** |
> | **ImageDoctor (Qwen2.5-VL)** | **0.00299** | 0.00003 | **0.225** | **2.863** | 0.108 |

---

> > ### Author Response · Authors · 2025-11-27
> > **Looking Forward to Your Feedback**
> >
> > Dear Reviewer LnNs,
> >
> > It seems we have not received your feedback on our response yet. As we enter the final week of the discussion phase, we would like to ensure that we have fully addressed your concerns.
> >
> > If there are any remaining questions or points that would benefit from further clarification, we would be grateful for the opportunity to elaborate.
> >
> > Thank you again for your time and consideration, and we look forward to your feedback.

---

### Author Response · Authors · 2025-11-23
**General Responses**

We sincerely thank all reviewers for their constructive feedback and are encouraged by the consistent recognition of our contributions. Reviewers highlighted that **ImageDoctor provides a unified and interpretable T2I evaluation framework**, offering multi-aspect scoring and pixel-level diagnostic heatmaps that address the limitations of traditional scalar metrics (**LnNs, yq1W, z1tc, bUcB**). They also noted that the **“look–think–predict” paradigm** is well-motivated and empirically validated through ablations (**bUcB**).

Reviewers further emphasized that our proposed **DenseFlow-GRPO** effectively leverages dense, heatmap-guided rewards, leading to **substantial improvements in T2I fine-tuning**, especially in correcting local artifacts and outperforming sparse-reward baselines (**yq1W, z1tc, bUcB**). They also acknowledged that ImageDoctor achieves **state-of-the-art correlation with human judgments** and provides interpretable, spatially grounded feedback valuable for both evaluation and reward modeling (**LnNs, yq1W, z1tc**).

We appreciate the reviewers’ positive comments on the **clarity of writing** and the **extensive experiments** demonstrating consistent improvements across datasets (**LnNs, yq1W**).

Below, we summarize the common issues raised across reviewers and the corresponding revisions made in the updated manuscript:

**1. Generalization and Choice of MLLM Backbones (LnNs, bUcB)**
**Response:** We have added new experiments using a different architecture and a smaller model, **InternVL3-1B**, showing that ImageDoctor with this new backbone still substantially outperforms the previous SOTA model **RichHF** across multiple dimensions (Tables R1–R4), particularly in semantic alignment.
We also evaluated **Gemini 3-Pro** and **GPT-5** on scalar prediction and found that generic MLLMs without task-specific fine-tuning perform significantly worse (Tables R8–R9).
These results demonstrate that ImageDoctor’s **training pipeline and grounded reasoning framework generalize well across model scales and architectures**, and further justify our use of open-source backbones, which uniquely allow multi-modal outputs and end-to-end fine-tuning.

**2. Trade-off Between Interpretability and Efficiency (LnNs, yq1W)**
**Response:** We added a dedicated discussion on the interpretability–efficiency trade-off and introduced a **fast inference mode (ImageDoctor-fast)** that skips explicit text reasoning during inference. This mode reduces inference latency by **7×** while preserving most of the performance (Tables R1–R6).
This flexible design ensures that ImageDoctor is practical for real-world deployment and offers a continuum between interpretability and efficiency, allowing users to choose the configuration most suited to their application needs.

**3. Generalization Beyond RichHF-18K (z1tc, bUcB)**
**Response:** We extended the evaluation to the **WISE benchmark**, using images generated by recent T2I models (SD3.5-medium, Flux-schnell, Kolors), which differ significantly from the training data. As shown in Figures 10–12 of the revised manuscript, ImageDoctor accurately identifies **physical inconsistencies, commonsense violations, cultural knowledge errors, and various model-specific artifacts**, demonstrating **strong diagnostic robustness far beyond the training distribution**.

Beyond these major concerns, we provide **point-by-point responses** to all remaining reviewer comments. All key revisions are highlighted in red in the updated manuscript. Prior to the score rollback, Reviewer z1tc had acknowledged that all concerns were satisfactorily addressed, while the other reviewers had not yet engaged further in the discussion or updated their scores.

We sincerely thank the new area chairs for their time and effort in handling our submission. We hope this concise summary helps quickly convey the paper’s contributions, revisions, and current review status.

---

### Meta-Review · Area_Chair_v6dq · 2026-01-14

**Summary:**

This paper introduces ImageDoctor, an evaluation framework for text-to-image generation that provides evaluatoin across four dimensions. It also provides pixel-level heatmap that highlights the misaligned regions. Reviewers gives scores of 6,6,8,6. Overall, the scores are positive.

The major concerns focus on the effectiveness of of MLLM Backbones, trade-offs between interpretability and efficiency, and the generalization beyond RichHF-18K. The rebuttal addresses most of the concerns with more experiment results and cleaner explanations.

**Reviewer Concerns:**

The concerns on the effectiveness of of MLLM Backbones, trade-offs between interpretability and efficiency, and the generalization beyond RichHF-18K have been well addressed.

**Reviewer Scores:**

Reviewers did not have follow up comments after the rebuttal.

---

### Decision · Program_Chairs · 2026-01-26

Accept (Poster)